# A CLOSED-LOOP VISUAL STIMULATION FRAMEWORK VIA EEG-BASED CONTROLLABLE GENERATION

## ABSTRACT

Recent advancements in artificial neural networks (ANNs) have greatly enhanced the ability to predict neural activity in response to visual stimuli. However, the inverse problem of designing visual stimuli to elicit specific neural responses remains challenging due to high experimental costs, the high dimensionality of stimuli, and incomplete understanding of neural selectivity. To address these limitations, we present a **closed-loop visual stimulation framework via electroencephalography (EEG)-based controllable generation**. It can iteratively generate the optimal visual stimuli to achieve the goal of controlling brain signals. This framework employs an EEG encoder, treated as a non-differentiable black-box model, to predict neural responses evoked by visual stimuli. By utilizing this encoder (or human experiment), we can quantify the similarity between the predicted (or recorded) neural responses and target neural states. Combining EEG feature extraction with a generation/retrieval module, the framework systematically explores large-scale natural image spaces to identify stimuli that optimally align with the desired brain state. Experimental results demonstrate that, irrespective of the precision of ANN-predicted brain activity, our framework efficiently converges to the theoretically optimal stimulus within a fixed number of iterations. Moreover, this framework generalizes effectively across diverse target neural activity patterns, underscoring its robustness and potential for broader applications in brain-inspired stimulus design. Our code is available at `https://anonymous.4open.science/status/closed-loop-F2E9`.

## 1 INTRODUCTION

The visual system exhibits selectivity, meaning different visual stimuli evoke distinct neural responses (Epstein & Kanwisher, 1998; Qiu et al., 2023). This property suggests that visual stimuli could, in principle, be designed to elicit specific neural responses, offering a novel, non-invasive approach to neuromodulation. Such neuromodulation technique offers several advantages: it is user-friendly, natural, and inherently well-aligned with human sensory processing. However, achieving precise neuromodulation through visual stimuli is highly challenging due to the high dimensionality of visual input space and our incomplete understanding of neuronal selectivity in visual system. Recent advances in controllable image generation techniques have enabled the creation of images with specific semantic attributes, typically conditioned on textual descriptions (Li et al., 2019; Epstein et al., 2023; Wei et al., 2024). While this represents a significant technological breakthrough, current methods lack the ability to conditionally generate stimuli based on neural states. To address this limitation, it is essential to develop frameworks capable of generating visual stimuli specifically optimized to modulate neural activity in a targeted manner, paving the way for more effective and precise neuromodulation through visual stimulation.

Many efforts have focused on precise control of brain activity through visual stimulation. For example, several works (Ponce et al., 2019; Walker et al., 2019; Bashivan et al., 2019) have sought to regulate neural activity at the neuronal level using targeted visual inputs. Notably, (Ponce et al., 2019) introduced a closed-loop experimental framework that integrates a deep generative neural network (GAN) with neurofeedback to iteratively generate images optimized to maximize the responses of specific neurons in the visual system. Despite their success in monkey experiments, these methods often lack generalizability and fail to capture the full diversity of visual features due to the small number of trials and constraints inherent in animal experiments. Moreover, they primar-

Figure 1: **Conceptualization.** The closed-loop visual stimulation framework includes three core components. (1) The *Black-box model* is used as a surrogate brain to generate neural responses to visual stimulation, and can be replaced by EEG data recorded from human participants in real closed-loop experiments. (2) The *Encoder* extracts the brain features associated with the target neural activity, which can be designed flexibly according to specific control goals. (3) The controllable image generator *Guided diffusion* synthesizes several candidate images. Through closed-loop iteration, the system continuously refines the visual stimulation to achieve the desired brain response.

ily focus on optimizing stimuli for individual neurons, which cannot reflect the complex, distributed neural coding patterns observed at a macroscopic scale, such as those captured in EEG signals. More recently, (Luo et al., 2024b) introduced the Visual Evoked Potential (VEP) Booster, a closed-loop framework designed to generate EEG biomarkers through visual stimulation. However, the VEP Booster primarily generates stroboscopic visual stimuli, rather than natural images that align with the known selectivity principles of the visual system (e.g., preferences for faces, objects, or semantic categories). Therefore, it is crucial for a closed-loop neuromodulation framework that uses natural image stimuli, capable of both flexibly controlling EEG signals and respecting the brain's inherent selectivity.

In this work, we develop a flexible closed-loop visual stimulation framework designed to achieve controllable EEG responses, as illustrated in Figure 1. By leveraging existing natural image datasets (Hebart et al., 2019) and pre-trained image generation models (Rombach et al., 2022), we utilize state-of-the-art diffusion models to identify fine-grained brain functional specializations in a data-driven manner. Our contributions are summarized as follows:

- We introduce a cutting-edge closed-loop visual neurofeedback framework that synthesizes natural images to control brain activity signatures. Our framework establishes a causal mapping between synthetic visual stimuli and specific EEG features in visual regions.

- By replacing traditional human EEG experiments with a black-box model (serving as a surrogate brain to predict neural responses to stimuli), we minimize dataset biases and enhance the model's ability to generalize to novel stimuli, providing valuable insights for future human subject experiments.

- We leverage state-of-the-art diffusion models to identify fine-grained visual selectivity, incorporating natural image priors to improve generalization. It allows for flexible design according to specific control goals, such as image retrieval to approximate neural activity generated by a reference image.

## 2 RELATED WORK

**Mapping Selectivity and Invariance from EEG.** Modern neuroscience posits that specific regions of the brain exhibit distinct sensitivities or preferences for particular types of stimuli (Tesileanu et al., 2022). This phenomenon, known as *selectivity*, describes how neurons or neural networks in these regions respond strongly to specific visual inputs. For instance, (Luo et al., 2024a) highlights

cases where neurons demonstrate pronounced selectivity for particular stimuli, underscoring their preference for specific visual features. In contrast, *invariance* refers to the brain's ability to respond consistently to distinct stimuli that convey the same information. In other words, different stimuli can elicit similar patterns of brain activity (Baroni et al., 2023). To explore the intrinsic invariance shared by ANNs and the brain, (Feather et al., 2023) proposed a method for generating model-equivalent stimuli, also known as model Metamers. Metamers evoke identical neuronal activations as a reference stimulus, providing a robust framework to examine the internal states of AI models and their alignment with neural processes. This approach provides critical insights into the shared computational principles underlying how artificial and biological systems process and represent information.

**Closed-loop Control of Brain Activity via Visual Stimulation.** Neuromodulation through visual stimulation holds significant promise for understanding neural mechanisms and developing treatments for various neurological disorders. For example, 40-Hz light flicker, which entrains gamma oscillations in the brain, has shown potential in treating Alzheimer's disease (Iaccarino et al., 2016; Martorell et al., 2019), while visual stimulation by natural images has been explored for improving mood in patients with depression and anxiety disorders (Mizumoto et al., 2024). A key approach in this field is the closed-loop control of brain activity, which allows for the real-time regulation of neural responses through continuous monitoring and feedback. Recent advances in generative models like GAN and diffusion have enabled the generation of optimal visual stimuli to achieve specific control of brain activity. For example, (Bashivan et al., 2019) applied gradient ascent to maximize the activity of the target neuron population with visual stimuli generated by a GAN-based image generator. Similarly, (Walker et al., 2019) proposed the "inception loops" paradigm, combining in vivo neural recordings with in silico modeling to synthesize visual stimuli that evoke desired neuronal responses. (Pierzchlewicz et al., 2024) developed a method to generate images using energy guidance to maximally activate neuronal responses in the V4 region of monkeys. More recently, (Luo et al., 2024b) employed a closed-loop strategy where a trained generative model iteratively refined VEP-EEG biomarkers. These advancements underscore the potential of closed-loop visual stimulation in precisely modulating brain activity.

**Brain-conditioned Controllable Image Generation** Traditional controllable image generation is typically conditioned on text, where the generation of images is guided by specific textual descriptions (Li et al., 2019; Epstein et al., 2023). In contrast, *brain-conditioned controllable image generation* directly uses the brain's neural activity, such as EEG, to guide the image generation process. A key technique in this field is the gradient-based method, which has become crucial for optimizing visual stimulus guided by brain activity (Luo et al., 2024b;a). This method involves iteratively refining visual stimuli by backpropagating the gradients of neural activity representations, allowing the brain states to be steered toward desired conditions or to achieve specific cognitive outcomes. This approach enables precise, adaptive stimulus optimization in response to real-time neural feedback, forming the foundation for personalized brain modulation. Recent advances have expanded the scope of gradient-based techniques by integrating more sophisticated neural encoding models and utilizing high-dimensional neural representations captured by various brain imaging modalities (Gu et al., 2023). These developments have significantly improved image generation, accounting for individual variability in neural responses. Moreover, the incorporation of deep learning models, such as guided diffusion models (Ye et al., 2023), has enabled the generation of highly detailed and context-specific stimuli, tailored to align closely with target neural states. These advancements represent a significant step forward in the field of brain-conditioned image generation.

## 3 METHOD

In this study, we develop a closed-loop framework to control brain activity through visual stimulation. The visual stimuli are generated by controllable generation models, conditioned on the EEG signals predicted by EEG encoder Figure 1. The framework is illustrated in Figure 2A. This closed-loop system is highly adaptable, allowing for the execution of various control objectives. For instance, by designing a control goal to minimize the distance between the EEG representation induced by the visual stimulus and a reference EEG representation (e.g., from a seen image), the system can perform a retrieval task (Figure 2B). Alternatively, by minimizing the distance in the power spectral density (PSD) features of the EEG, the system can implement a EEG-conditioned

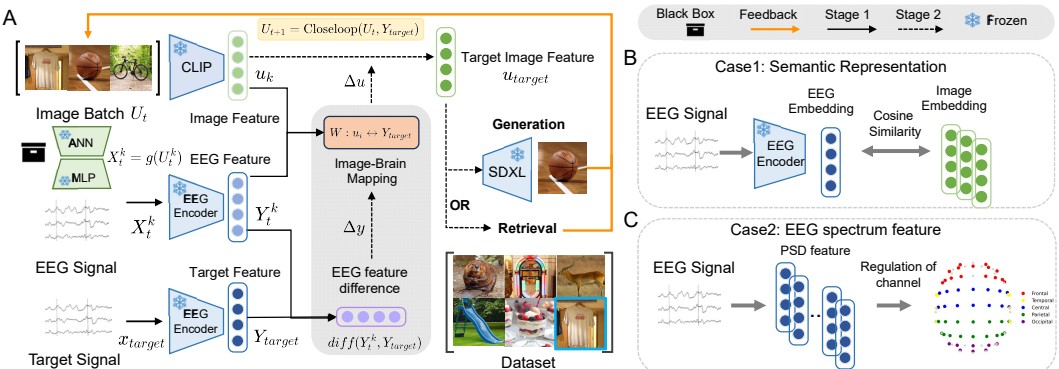

Figure 2: **Closed-loop visual stimulation framework via EEG-based controllable generation.**
(A) We employs a closed-loop iterative process to approximate neural representations derived from
EEG signals $X$. The encoding model $g$, which maps images to synthetic EEG, is designed as a black-
box model to broadly simulate the process of regulating brain responses $Y$. The EEG Encoder $f$ is
tailored to accommodate various neural features $U$. The image with a higher brain similarity score
$sim\langle u_j, u_{\text{target}} \rangle$ is retained and passed back to the image generator to generate optimized stimuli
with a natural image. (B) Example of semantic feature extraction from a pre-trained EEG encoder
$f$, aligned with CLIP embedding. In this case, our algorithm performs a retrieval task to identify
the optimal image $u_i$ that best matches the $u_{\text{target}}$. (C) Channel-wise energy feature using Power
Spectral Density (PSD) features. Generative models are iteratively applied to modify the images.
For more details, refer to Section 3.1.

generation task (Figure 2C). To support these tasks, we design two distinct feature extractors: one
for retrieval and one for generation. If the system begins to favor images with specific colors or
textures, it will recognize the relevance of these features to the target class and assign them higher
weight in subsequent iterations. Through this closed-loop iterative process, the system can continu-
ally optimize the visual stimulus to better elicit the desired EEG responses.

## 3.1 CLOSED-LOOP FRAMEWORK

We formulate the EEG signals as $X \in \mathbb{R}^{C \times T}$, where $C$ is the number of EEG channels and $T$
represents the length of data points. The image set, containing $N$ images, is denoted as $\Omega$, with
each image labeled sequentially with $1, 2, ..., N$ for simplicity. Concurrently, we use the encoding
model $g$ to predict brain activity signal $X = g(U) \in \mathbb{R}^{N \times C \times T}$. Our objective is to derive brain
activity embeddings $Y = f(g(U)) \in \mathbb{R}^{N \times F}$ from the images $I \in \mathbb{R}^{N \times 3 \times H \times W}$, where $f$ is the
feature mapping function from $X$ to $Y$, $U$ is the set of stimulus images set, and $F$ represents the
dimension of embedding. Our iteration process can be approximated as a value-based iterative
Markov Decision Process (MDP). The state is represented as the probability distribution of each
image $P(u)$ in the image database belonging to target category $u_{target}$. The state updated after each
iteration corresponds to a state transition in the MDP. In each iteration, the framework determines
which image to select, represented as an action in the MDP. In our model, let $j \in [\![1, N]\!]$, the reward
is defined as the similarity score between the selected or generated image $u_i$ from database and the
features of the target category $u_{\text{target}}$:

$$sim\langle u_j, u_{\text{target}} \rangle = \frac{f\left(g(u_j)\right) \cdot f\left(g(u_{\text{target}})\right)}{\|f\left(g(u_j)\right)\| \|f\left(g(u_{\text{target}})\right)\|} \tag{1}$$

Let $u_i$ be any image in the search space, which is the target of model evaluation. During the iteration
of the $t$ to $t+1$ step, we update $S_{t+1}(u_i)$ based on $u_i$. The weight coefficient $\alpha$ controls the
cumulative probability increment. Let $u_+$ be the image that the system considers to be closest to the
target category by computing EEG feature similarity. For the history subset $H$ of selected images $k$,
the posterior probability that $u_i$ is the most similar to the target image is updated as follows:

$$S_{t+1}(u_i) = \alpha \cdot S_t(u_i) + (1 - \alpha) \cdot \frac{\exp\left(s(u_+, u_i)\right)}{\sum_{k=1}^{H} \exp\left(s(u_+^k, u_i)\right)} \cdot S_t(u_i) \tag{2}$$

where $s$ is the cosine similarity of CLIP (Radford et al., 2021) embedding. The update probability $P_{t+1}(u_i)$ for $u_i$ is computed by normalizing the exponentiated value of the updated score $S_{t+1}(u_i)$ over the sum of exponentiated scores for all $u_j$ in the dataset, ensuring that the probabilities across all $u_i$ sum to 1:

$$P_{t+1}(u_i) = \frac{\exp\left(S_{t+1}(u_i)\right)}{\sum_{j=1}^{N} \exp\left(S_{t+1}(u_j)\right)} \tag{3}$$

In step $t$ iteration, our framework operates as follows. First, we initialize a set of random images $U_0 = \{u_1, u_2, \ldots, u_j\}$. Using the pretrained encoding model $g$ to synthesize EEG signals $X_i$ from these stimuli. Second, for any given representation function $Y_i$, we calculate the neural activity representation $Y_i = f(g(U_i)) \in \mathbb{R}^{N \times F}$ from the predicted signal $x_i$, to estimate the difference based on the target neural representation $Y_{target}$. Third, the similarity score $sim\langle u_j, u_{\text{target}}\rangle$ between each neural representation derived from each current stimulus $u_j$ and the target representation is computed. Subsequently, stimulus images exhibiting higher similarity scores are more likely to be selected. Based on $sim\langle u_j, u_{\text{target}}\rangle$, stimulation is probabilistically sampled, favoring images that are closer to the target representation. Finally, the sampled images are used to retrieve similar images for the step $t + 1$ or input into the diffusion model to generate new stimulus samples.

## 3.2 BLACK-BOX ENCODING MODEL

Instead of recording real EEG data, we employ a pre-trained EEG encoder $g_\theta$, treated as a black-box model, to map an image $I_i \in \mathbb{R}^{3 \times H \times W}$ to a synthetic EEG $X_i$. This model predicts the EEG responses corresponding to the visual stimulus. The predicted EEG response can be substituted with actual EEG recordings obtained from human participants during experimental settings. The EEG encoder involves a pre-trained image feature extractor to obtain image embedding aligned with EEG embedding, and a regression model to generate EEG signals from the embedding representation. To test the robustness and generalizability of EEG encoder, we implement two CNN models as image feature extractors, including AlexNet (Krizhevsky, 2014) and CORnet-S (Kubilius et al., 2019). We then train regression models, denoted as $\hat{X}$, to predict the neural response according to the image features using supervised learning with the ground truth EEG (from image-EEG paired data).

In the encoding model, we modify the output layer of the CNN, replacing its 1000-neuron configuration with a $C \times T$-neuron layer, where each neuron corresponds to one of the flattened EEG data ponits $C \times T$. Each subject is associated with unique model parameters, which are obtained via pretrained models, applied across all EEG time points $T$. Given the input training images $I$ and their corresponding target EEG data $\hat{X}$, the model updates its weights by minimizing the mean squared error (MSE) between predicted EEG $X$ and the target EEG $\hat{X}$. This setup ensures a personalized and accurate prediction of synthetic neural activity. This framework ensures a personalized and precise prediction of synthetic neural activity.

## 3.3 INTERACTIVE SEARCH

To identify the optimal stimulus that elicits the desired neural activity, we search for images that generate EEG features similar to the target. The target query image is unknown, and the corresponding EEG feature is observable. To address the challenge of initiating retrieval without a clear query image, we use the mathematical framework of (Ferecatu & Geman, 2007), based on mind matching. It begins with a random sample of images, and through iterative steps, the user selects the image that most closely aligns with the intended category. In our case, this process is adapted to match the target neural feature. The detailed algorithmic procedure is outlined in Algorithm 1, which effectively identifies an optimal subset of images that maximizes the similarity score with respect to the target EEG feature.

In our framework, the *Closed-loop Retrieval Iteration Algorithm* functions as a sequence of state transitions aimed at maximizing the similarity between the current neural feature and the target. The process begins with a randomly selected set of images $U_0$, without prior knowledge of the specific features of the target image. We use a roulette wheel selection algorithm to choose from current images based on the similarity measure $sim\langle u_j, u_{\text{target}}\rangle$. The system updates the probability $p_t(u_j)$ for each image in the database belonging to the target class, based on the response model's prediction $Y = f(g(U)) \in \mathbb{R}^{N \times F}$. Subsequently, the system calculates the distance between the

---

**Algorithm 1** Closed-loop Retrieval Iteration Algorithm

---

1: **Initialize:** Set initial set $U_0 = \{u_1, u_2, \ldots, u_k\}$, where $U_0 \subseteq \Omega$.
2: **repeat**
3:     **Action Selection:** $U_t = \{u_1, u_2, \ldots, u_k\}$ from $\Omega$ based on $p_t(u)$.
4:     **Reward Calculation:**
$$sim_{\max} = \max\ sim\langle u_k, u_{\text{target}}\rangle$$
5:     **if** $sim_{\max} < threshold_1$:
6:        Go to Step 3.
7:     **else**:
8:        **Optimal Action Reference:**

$$\{u_{\text{top1}}, u_{\text{top2}}\} = \arg \max_{\substack{u_k \in U_t \\ \text{top 2}}} \frac{\exp\left(sim\langle u_k, u_{\text{target}}\rangle\right)}{\sum_{u_h \in H} \exp\left(sim\langle u_h, u_{\text{target}}\rangle\right) + \sum_{u_k \in U_t} \exp\left(sim\langle u_k, u_{\text{target}}\rangle\right)}$$

9:        **if** $sim\langle u_{target}, u_{top1}\rangle$ or $sim\langle u_{target}, u_{top2}\rangle > threshold_2$:
10:       **CLIP-based Retrieval:** Using $u_{\text{top1}}$ and $u_{\text{top2}}$, retrieve the top-$k$ images $\{u'_1, u'_2, \ldots, u'_k\}$ from $\Omega$ that have the highest similarity $s$:

$$u'_k = \arg \max_{u \in U} \{s(u, u_{\text{top1}}), s(u, u_{\text{top2}})\}.$$

11:       **Update Action Set:** Update the subset $U_{t+1}$:

$$U_{t+1} = \{u'_1, u'_2, \ldots, u'_k\}.$$

12:       **Recurse on** $U_{i+1}$: Repeat the process for the new action set $U_{t+1}$, treating it as the current action set $U_t$ for the next iteration.
13: **until** $s_{\max} \geq threshold_{primary}$
14: **Return:** Return the best action set $U_t$ as the final set of retrieved images.

---

brain activity feature vector of the target image and the brain activity feature vector predicted by the image selected by the roulette wheel algorithm (i.e., the image deemed to be closest to the target class). Once an image is identified as the best in a given iteration, the likelihood of similar images in the search space belonging to the target class is increased. See more implementation details in Appendix A.1.2.

### 3.4 HEURISTIC GENERATION

Retrieving the optimal image stimulus solely within the image feature space restricts the ability to closely align with the target brain activity. To design an optimal stimulus with greater precision, we employ StableDiffusion XL-turbo for image-guided optimal stimulus generation. The pretrained guided diffusion model $G(U_t)$ generates new visual stimuli through an image-to-image process. Based on MDP, we integrate a genetic algorithm to guide the generator in producing images that align with the target neural activity while maintaining global optimality. The specific procedural steps of our algorithm are outlined in Algorithm 2. Unlike the retrieval process in Algorithm 1, after sampling the stimulus image in each roulette step, we perform feature crossover on the image and randomly sample new images from the image space. Mutation is performed based on the current images features $U_t$. See Appendix A.1.3 for additional details on the evolution process. Throughout this process, the relative order of original CLIP features is preserved in each sample to ensure that semantically coherent images, which are understandable to humans after mutation.

## 4 EXPERIMENTS

### 4.1 SETUP

**Encoding Model** We conducted our experiments using the training set of the THINGS-EEG2 dataset (Gifford et al., 2022; Grootswagers et al., 2022). For further details on the dataset, please refer to Appendix A.1.1. During the training phase, we employed a batch size of 64 images and

---

**Algorithm 2** Closed-loop Generative Iteration Algorithm

---

1: **Initialize:** Set initial set $U_0 = \{u_1, u_2, \ldots, u_k\}$, where $U_0 \subseteq \Omega$.
2: **repeat**
3:     **Selection:** $U_t = \{u_1, u_2, \ldots, u_k\}$ from $\Omega$ based on $p_t(u)$.
4:     **Sampling:** Based on the calculated similarity scores, sample from $U_t$ using:

$$P(u_k) = \frac{\exp\left(sim\langle u_k, u_{\text{target}}\rangle\right)}{\sum_{u_{k'} \in U_t} \exp\left(sim\langle u_{k'}, u_{\text{target}}\rangle\right)}$$

    where $P(u_k)$ is the sampling probability for each $u_k \in U_t$.
5:     **Crossover:** Draw two distinct samples $u_a, u_b$ from $U_t$ based on $P(u_k)$, and output new
    samples by combining the partial embedding of $u_a$ and $u_b$:

$$F(u_{\text{tmp}}^{(1)}) \leftarrow \alpha \cdot F(u_a) + (1 - \alpha) \cdot F(u_b)$$

$$F(u_{\text{tmp}}^{(2)}) \leftarrow \alpha \cdot F(u_b) + (1 - \alpha) \cdot F(u_a)$$

    where $\alpha$ is a crossover control factor.
6:     **Mutation:** Based on $P(u_k)$, apply mutation to the drawn images $u_{\text{c}}$ from $U_t$, and another
    image $u_{\text{d}}$ is drawn from the remaining $U_t$ (i.e., $U_t \setminus \{u_{\text{c}}\}$):

$$F(u_{\text{tmp}}^{(3)}) \leftarrow \beta \cdot F(u_c) + (1 - \beta) \cdot F(u_d)$$

    where $\beta$ is a mutation control factor.
7:     **Generation:** Generate a new set of images $U_{\text{gen}} = \{u_{\text{gen}}^{(1)}, u_{\text{gen}}^{(2)}, u_{\text{gen}}^{(3)}\}$ according to the out-
    puts of crossover and mutation phase.
8:     **Selection:** Combine $U_{\text{gen}}$ with $U_t$ and randomly selected samples $U_{\text{random}} =$
    $\{u_{\text{ran}}^{(1)}, u_{\text{ran}^{(2)}}, \ldots, u_{\text{ran}}^{(n)}\}$, where $U_0 \subseteq \Omega$.
9:     **Update Action Set:** Update the subset $U_{t+1}$:

$$U_{t+1} \leftarrow \{U_t, U_{\text{gen}}, U_{\text{random}}\}$$

10:    Replace the old population with the new set of images $U_{i+1}$.
11: **until** similarity score converges or reach the maximum number of cycles.

---

utilized the Adam optimizer with a learning rate of $10^{-5}$, a weight decay parameter of 0, and default values for the other hyperparameters. Training was conducted over 50 epochs, with EEG responses for test image conditions synthesized using the model weights from the epoch that yielded the lowest validation loss. For each participant, the models generated EEG signals with a shape of 17 EEG channels $\times$ 250 EEG time points as the output corresponding to the input images. All experiments were conducted on a single NVIDIA 4090 GPU. For additional training details and validation procedures, see Appendix A.3.

**Target Features of EEG** We designed different target EEG features for semantic feature and spectral signature case. In the retrieval task based on semantic representation, the system randomly selects target images from the test set of THINGS-EEG2, with an index greater than 12 in each class. These selected images are excluded from the retrieval space of $200 \times 12 = 2400$ images. In the generation task based on spectral features, in order to ensure that the regulation is meaningful, we calculated the EEG feature similarity matrix corresponding to the prediction of the $200 \times 1$ image from the test set, and took the top-3 images with the lowest similarity in each class after row averaging as the target for testing. We use the pre-trained encoding model (AlexNet, CORnet-S) and pre-trained EEG encoders (ATM-S (Li et al., 2024), PSD) to process the target images and extract their corresponding EEG features.

### 4.2 REGULATION OF BRAIN SEMANTIC REPRESENTATION

To evaluate the effectiveness of our framework in achieving the target neural activity representation, we conducted a retrieval task in the image space. We treated the encoding model $g$ as a black-box model, ensuring that gradients were not used to update its parameters. This approach allowed us

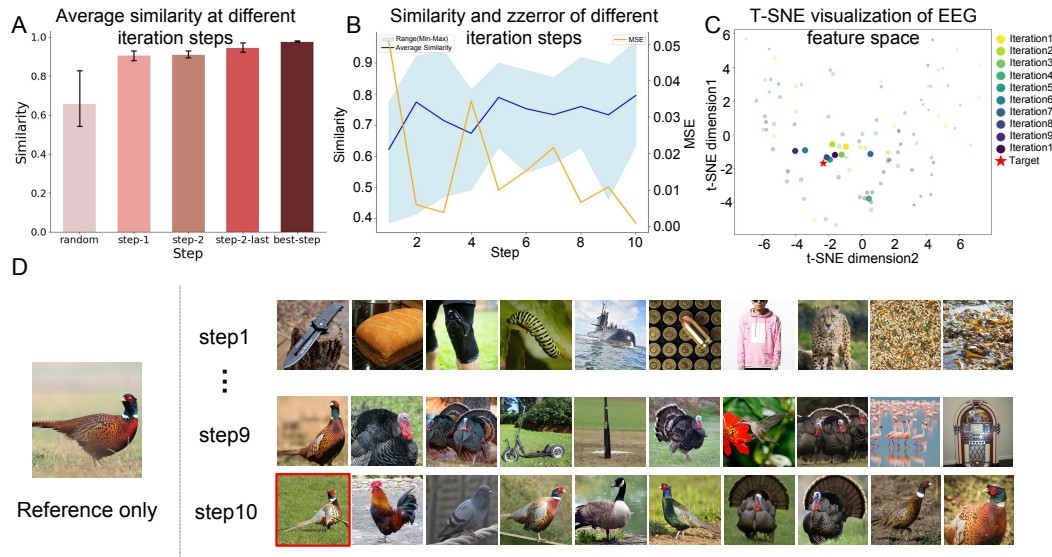

Figure 3: **Results of our framework in the retrieval task.** (A) Similarity between the neural representation obtained by our framework at different iteration steps (i.e., step-1, step-2, step-2-last, best-step) and the target neural representation compared to random stimulus (i.e., random). (B) The evolution of EEG representation similarity (blue) and loss curves (yellow) on Subject 8 at different iteration steps. (C) The t-SNE visualization of Subject 8's latent trajectories within the feature space across all iterations. (D) The images retrieved by our framework at different iteration steps. Only the neural activity representation evoked by the reference image is known during the iteration process. See Appendix A.4 for more results.

to focus on the closed-loop regulation framework itself. The retrieval task was performed on the test set of the THINGS-EEG2 dataset, which consists of 2400 images. We used the EEG encoder ATM-S to obtain EEG semantic representations aligned with $1 \times 1024$ CLIP image features. Before initiating the retrieval, random initialization was used to scatter 10 initial points as widely as possible in the image feature space. During the search process, each initial image sample calculates its cosine similarity with the global image features, and cumulative probability is applied to increase the likelihood of selecting new images that bring the EEG representation closer to the target. In the image feature space, the initial sample points expand iteratively, forming a small region, and gradually converge toward the theoretically optimal stimulus image. The termination condition for iterations is the similarity $s(u_+, u_i) > threshold_{primary}$.

In Figure 3, we report our retrieval results based on EEG semantic representation. In Figure 3A, we show the similarity scores of stimuli compared to random stimuli at different time steps during the iteration process. Figure 3B displays the average similarity and mean squared error between the predicted and expected EEG features at various iteration time points for subject 8. Figure 3C illustrates the convergence patterns from initial to final positions for selected iterations (e.g., iterations 1 and 10) across multiple cycles. In each iteration, ten images are presented, with points representing the closest match to the target stimulus at each step. Notably, these points gradually move toward the target stimulus, marked by a red pentagram, across successive iterations. For a given target neural activity representation, our framework iteratively predicts intermediate EEG results and retrieves stimulus images at each iteration. Importantly, only the neural activity representation evoked by the reference image is known throughout this process. Through successive iterations in Figure 3D, the framework refines its selection and ultimately retrieves an image (outlined in red) that closely matches the semantic representation of the reference image.

### 4.3 REGULATION OF INTENSITY OF NEURAL ACTIVITY

We implemented a closed-loop stimulus image generation framework using the $200 \times 1 = 200$ image space of THINGS-EEG2 as initialization. We set the crossover rate $\alpha$ to 0.6, the mutation rate $\beta$ to

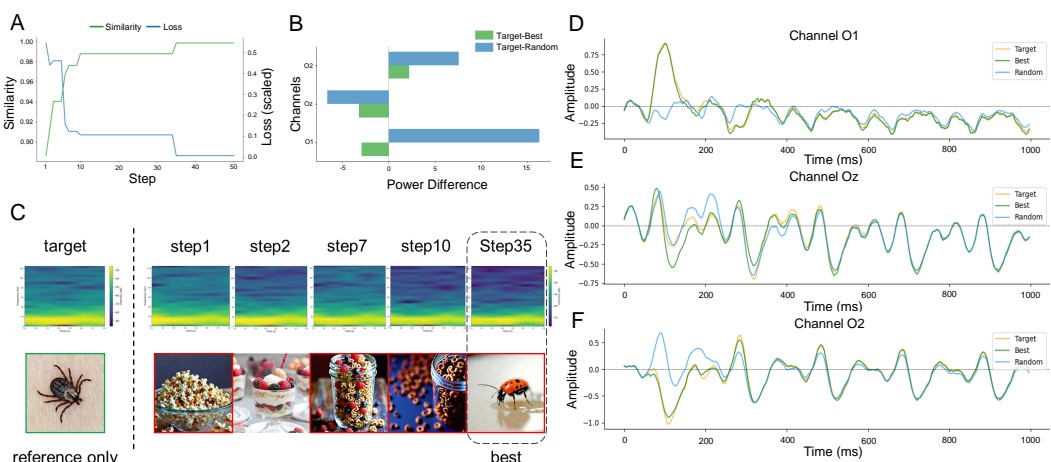

Figure 4: **Results of our framework in the generation task.** (A) Similarity and loss curves of EEG neural representations for Subject 8. (B) The difference of PSD between the neural activity representations evoked by the final step of generated and random stimulus, with the target neural representations used as the relative baseline. (C) For a given target EEG semantic representation, our framework iteratively predicts synthetic data, extract feature and synthesizes images at each iteration. The image enclosed by a red border represents the image synthesized by the generator, while the unbordered image is a sample selected from the original dataset. See Appendix A.5 for more generated examples. (D) EEG timing diagram generated by our stimulus images for $O_1$ channel. (E) EEG timing diagram generated by our stimulus images for $O_z$ channel. (F) EEG timing diagram generated by our stimulus images for $O_2$ channel.

0.2, and randomly select 10 images from 200 images during initialization. We used StableDiffusion XL-turbo (Rombach et al., 2022) integrated by IP-Adapter (Ye et al., 2023) to generate new samples each time based on the new stimulus images obtained after crossover and mutation, and randomly selected 2 samples from the image feature space, calculated the similarity of EEG activity representation, and selected the next step of stimulation according to the roulette method of cumulative probability.

The results of our stimulus generation experiments are shown in Figure 4. Figure 4(A) shows the similarity and mean square error between the EEG features generated by the step stimulation image at different iterations and the target EEG features. In addition, we calculated the explained variance of different channels and selected the three channels $O_1$, $O_z$, and $O_2$ with the largest variance for regulation. Figure 4(B) shows the comparison of the PSD of the EEG predicted by the random and step-best samples relative to the target EEG representation. Figure 4(DEF) plots the synthetic EEG of three different channels obtained by step-best, random and target stimulation images respectively. All three channels show that the EEG corresponding to step-best, random and target images is quite different before 100 data points (corresponding to 0.4s). After 0.4s, due to the limitations of the encoding model itself, the synthetic EEG of the target image is not much different from the synthetic EEG of the optimal stimulation and the synthetic EEG of the random image. This corresponds to Fig.4 in (Gifford et al., 2022). Using the tick image as an example, Figure 4(C) shows the image and its corresponding time-frequency features, as well as the generated image and corresponding features at each iteration.

## 4.4 REGULATION OF INDIVIDUAL VARIABILITY

Table 1 summarizes the results in the retrieval setting (corresponding to the representation score, SS) and the generation model setting (corresponding to the intensity score, IS), highlighting the results of our framework in achieving the optimal number of iterations in a given search space. The data show that for different target EEG features, our method has a good improvement in feature similarity across different subjects. For instance, the similarity score (SS) of the semantic feature of

Subject 7 is improved from 0.874 in step-1 to 0.974, with an improvement of 10.04%. Similarly, the feature similarity score (IS) of the channel intensity of Subject 8 is improved from 0.913 in step-1 to 0.990, accompanied by a 7.744% improvement. Even on the subjects with poor performance, our framework achieves a positive performance, which shows that our framework has a generalized improvement effect across different subjects, highlighting its potential in practical applications. See Appendix A.2 for more detailed quantitative results.

Table 1: **Performance (EEG semantic representation and intensity) of brain responses.** We provide two metrics: EEG semantic representation score (i.e., SS) and EEG response intensity score (i.e., IS) to measure the difference between the neural activity generated by the optimal stimulation image we obtained and the target EEG neural activity.

| Subject | Step-1 | | Step-Best | | Improvement | |
|---|---|---|---|---|---|---|
| | SS | IS | SS | IS | $\Delta$SS (%) | $\Delta$IS (%) |
| 1 | 0.871 | 0.989 | 0.967 | 0.997 | 9.593 | 0.801 |
| 7 | 0.874 | 0.960 | 0.974 | 0.995 | 10.040 | 3.444 |
| 8 | 0.904 | 0.913 | 0.976 | 0.990 | 7.162 | 7.744 |
| 10 | 0.915 | 0.986 | 0.961 | 0.998 | 4.587 | 1.163 |

## 5 DISCUSSION AND CONCLUSION

In this study, we developed a flexible closed-loop visual stimulation framework for controlling EEG signatures. To the best of our knowledge, this is the first work to successfully employ closed-loop generation of natural images to modulate brain activity.

**Technical Impact:** Our framework demonstrated the potential of flexibly controlling EEG signals through visual stimulation. We employed a closed-loop iterative strategy, where new random stimuli are sampled each time a new round of stimulus images is generated. The gradient of the EEG objective is passed to the diffusion model in a proxy manner, eliminating the need for training or updating the weights of the generative model. This approach demonstrates that our framework is an efficient and optimal closed-loop stimulus generation method, capable of achieving the desired neural modulation without requiring any model parameter updates. It opens new avenues for applications in brain-computer interfaces, neuro-feedback systems, and therapeutic interventions for neurological disorders that require precise regulation of brain activity (Jang et al., 2021; Alamia et al., 2023).

**Neuroscience Insights:** Our study provides valuable insights into the neural mechanisms underlying visual perception and stimulus processing. First, we demonstrated the successful modulation of activity in specific electrode channels, indicating that neural activity in targeted brain regions can be fine-tuned through controlled visual stimulation. Second, we showcased our framework's ability to guide the brain in generating specific neural representations, which is crucial for understanding how different brain regions process visual information and respond to external stimuli. Furthermore, our framework establishes a causal link between visual stimuli and neural responses. By connecting specific EEG patterns to visual representations, our work deepens the understanding of how neural signatures correlate with perceptual experiences.

**Interesting Phenomena and Future Directions:** Our findings demonstrate that different stimulus images in our framework can produce similar or identical EEG features, confirming the existence of Metamers (Feather et al., 2023) and suggesting that Metamers are not necessarily unique. The presence of multiple Metamers highlights the ill-posed nature of generating visual stimuli conditioned on EEG features. Future research should focus on understanding the neural mechanisms that lead to the generation of similar EEG features from different stimuli. Another promising direction is the integration of more sophisticated models that account for inter-individual variability in neural responses, aiming to fine-tune the stimulus generation process for personalized neuromodulation and enhanced brain-computer interaction (Alamia et al., 2021). Further exploration could involve integrating this closed-loop framework with other brain imaging modalities, such as fMRI or MEG. Additionally, it is crucial to formulate control goals aimed at regulating specific EEG characteristics to modulate brain functions, such as a control objective on EEG features for emotion regulation.

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

# A APPENDIX

## A.1 MORE IMPLEMENTATION DETAILS

### A.1.1 DATASETS

We conducted our experiments using the training set of the THINGS-EEG2 dataset (Gifford et al., 2022; Grootswagers et al., 2022), which consists of a large EEG corpus from 10 human subjects performing a visual task. The experiments used the Rapid Serial Visual Presentation (RSVP) paradigm for orthogonal target detection tasks to ensure participants' attention to the visual stimuli. All 10 participants underwent 4 equivalent experiments, resulting in 10 datasets with 16,540 unique training image conditions, each repeated 4 times, and 200 unique testing image conditions, each repeated 80 times. In total, this yielded (16,540 training image conditions × 4 repetitions) + (200 testing image conditions × 80 repetitions) = 82,160 image trials. The original data were recorded using a 64-channel EEG system with a 1000 Hz sampling rate. For preprocessing, the data were first downsampled to 250 Hz and 17 channels were selected from the occipital and parietal regions, which are closely related to the visual system. The EEG data were then segmented into trials, spanning from 0 to 1000 ms post-stimulus onset, with baseline correction applied using the mean of the 200 ms pre-stimulus period. Multivariate noise normalization was applied to the training data (Guggenmos et al., 2018).

### A.1.2 RETRIEVAL PIPELINE

We provide a more detailed description of algorithm 1. The algorithm begins by initializing equal selection probabilities for each image in the candidate set, denoted as $p_0(u) = \frac{1}{N}$, where $N$ is the total number of images in the retrieval set. This initialization with equal probabilities reflects the absence of prior information, serving as an exploratory phase. In each iteration (representing a **state** in the MDP framework), a subset of images $U_t = \{u_1, u_2, \ldots, u_j\}$ is selected from the candidate images set $U$ based on the current selection probabilities $p_t(u)$.

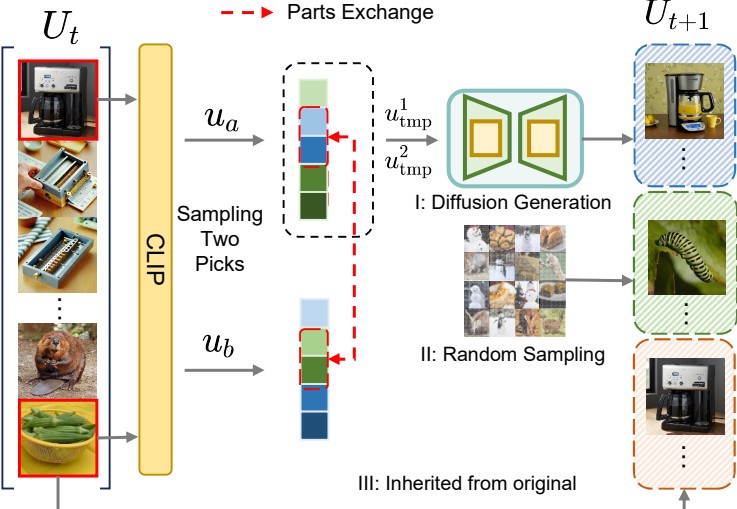

Figure A.1: Generating subsequent images based on the current round is achieved through crossover, variation, and a guided diffusion model. Both crossover and mutation operations preserve the relative ordering of CLIP features, thereby maintaining their semantic coherence.

For each image $u_j$ in the subset $U_t$ the algorithm computes a similarity score $sim\langle u_j, u_{\text{target}}\rangle$ by comparing the image's representation with the target. This similarity score acts as an immediate **reward** within the MDP framework. The maximum similarity score among the subset is identified as a measure of the effectiveness of the current action. If $sim_{\max}$ does not meet a predefined $threshold_1$, the reward is considered insufficient, and the algorithm returns to the image selection step, effectively trying a new action within the same state. If $sim_{\max}$ meets or exceeds the threshold, the algorithm proceeds to identify the two images $u_{top1}$ and $u_{top2}$ with the highest similarity

scores. These two images act as reference points for updating the probabilities of other images in the subsequent state.

As for each image $u_j$ in $U$ that surpasses $threshold_2$ with either $u_{top1}$ or $u_{top2}$, its selection probability $P_{t+1}(u_j)$ is updated by multiplying with a constant factor, representing a policy improvement step that prioritizes images likely to yield higher rewards. After updating, a Softmax function is applied to normalize the probabilities, focusing selection weight on images more similar to the target. This normalization step reflects the transition to a new state with an updated policy. The iteration continues, with the algorithm transitioning through states by selecting new subsets based on the refined probabilities, until $sim_{\max}$ reaches $threshold_{primary}$. At this point, the loop terminates, as the algorithm has successfully identified an optimal subset of images that maximizes the similarity reward to the target.

### A.1.3 GENERATION PIPELINE

We provide a more detailed description of algorithm 2. As illustrated in Figure A.1, each image set consists of three parts:

- Part one: This step focuses on the diffusion generation process. From the image set of last iteration, two images, denoted as $u_a$ and $u_b$, are sampled using a roulette wheel selection method. A random crossover is then applied to part of their image embeddings, with the crossover starting at a different index each time. The newly combined image embedding is then used as input to the diffusion process. This increases the variability of the image set while preserving the high-quality components of the image embeddings.

- Part two: In this step, images are randomly sampled from the original image dataset, excluding those that have already been selected in earlier iterations. This ensures that the new image set introduces novel elements while avoiding repetition.

- Part three: This part inherits the image $u_a$ and $u_b$.

By combining these three parts, we obtain a new image set for the next iteration.

## A.2 Additional Quantitative Results

### A.2.1 Iteration Improvement From Different subjects

Based on the conclusions drawn from Figure A.4, we employ the pre-trained AlexNet end-to-end model as the EEG encoder and use ATM-S, which is based on S-S (both the training and testing signals are synthesized), to obtain semantic representations aligned with 1×1024 CLIP image features. The experimental design involves randomly selecting 50 categories, resulting in a retrieval space of 50 × 12 = 600 images. Specifically, we present the iterative performance improvements for three different targets randomly selected from the test set, with results reported for Subjects 1, 7, 8, and 10. As shown in Figure A.2, we calculate the EEG feature similarity of Subject 1, 7, 8, and 10 at random, step-1, and step-best in the iterative process respectively.

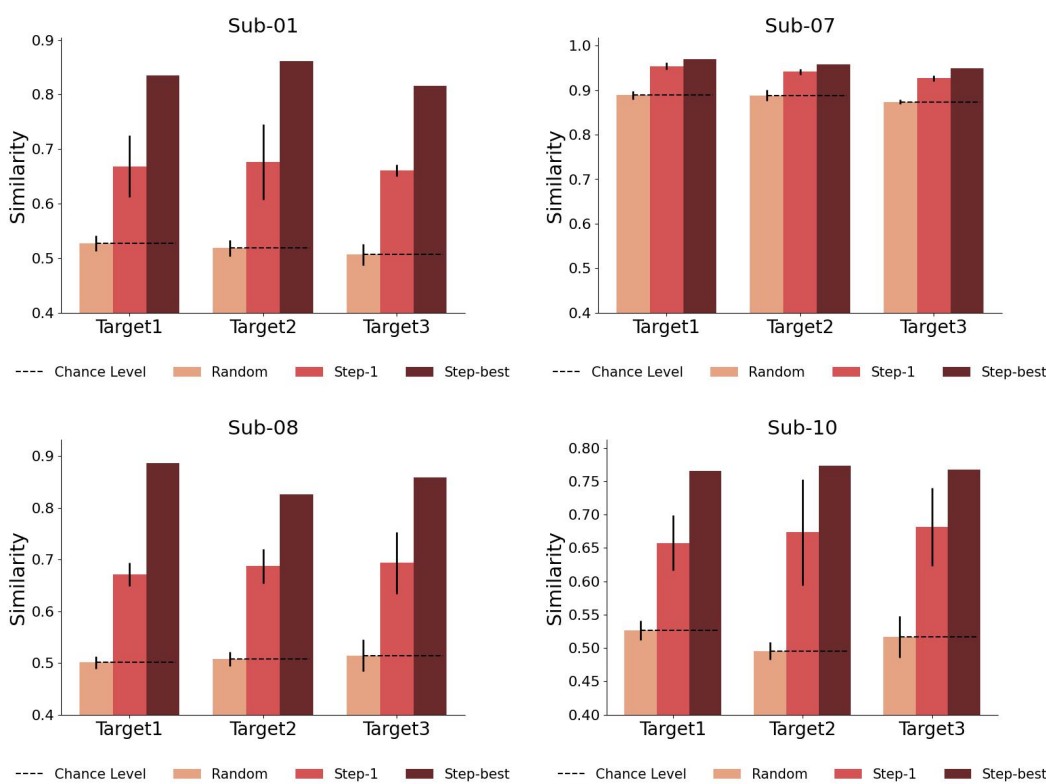

Figure A.2: **Comparison of improved performance by different targets.** We present the similarity scores of EEG features generated by random stimulation, open-loop stimulation (step 1), and step-best stimulation, in comparison to the target features. Each subject randomly selected 3 images from the retrieval space as target images.

### A.2.2 Performance of Different Target Images Across Subjects

We report the results of iterative optimization using different targets in two different cases. The results for each subject are shown, along with the average percentage improvement across 5 random seeds. For the semantic feature case, unlike the setting in Table 1 of the main text, which uses real EEG for training and performs retrieval on synthetic EEG, we determined that training and testing with synthetic EEG yielded the highest accuracy based on the retrieval performance shown in Figure A.4. As a result, we retrained each subject and summarized the results in Table A.1. For the intensity feature case, we selected 3 images using the method described in Section 4 and supplemented the iterative improvement performance. We performed t-tests on EEG semantic and spectral features across all subjects to assess the efficacy of our proposed method. Additionally, we performed correlation analyses to investigate the relationships between semantic features and clip representation, as well as between PSD feature and clip representation, as shown in Figure A.3.

Table A.1: **Performance (EEG semantic representation and intensity) of brain responses.** We provide two metrics: EEG semantic representation score (i.e., SS) and EEG response intensity score (i.e., IS) to quantify the similarity of generated EEG and target EEG. The table below records the SS & IS values for each subject, showing the SS & IS value from the first round of stimulation, the SS & IS value achieved after multiple rounds of closed-loop control (the optimal result), and the improvement in control. All these results are calculated from pretrained AlexNet models.

| | Random | | Step-1 | | Step-Best | | Improvement | |
|---|---|---|---|---|---|---|---|---|
| Subject | SS | IS | SS | IS | SS | IS | $\Delta$SS (%) | $\Delta$IS (%) |
| 1 | 0.5174 | 0.9632 | 0.6686 | 0.9729 | 0.8375 | 0.9976 | 16.8859 | 2.4790 |
| 2 | 0.5197 | 0.9678 | 0.6675 | 0.9764 | 0.7372 | 0.9998 | 6.9701 | 2.3406 |
| 3 | 0.5113 | 0.9883 | 0.6597 | 0.9927 | 0.7871 | 0.9980 | 12.7402 | 0.5306 |
| 4 | 0.5065 | 0.9650 | 0.6498 | 0.9836 | 0.8299 | 0.9963 | **18.0136** | 1.2690 |
| 5 | 0.5315 | 0.9788 | 0.6937 | 0.9768 | 0.8418 | 0.9979 | 14.8151 | 2.1055 |
| 6 | 0.6747 | 0.9836 | 0.8099 | 0.9856 | 0.8826 | 0.9961 | 7.2634 | 1.0461 |
| 7 | 0.8838 | 0.8955 | 0.9410 | 0.9033 | 0.950 | 0.9742 | 1.8237 | **7.0879** |
| 8 | 0.5077 | 0.8344 | 0.6838 | 0.9435 | 0.8568 | 0.9925 | 17.3066 | 4.8947 |
| 9 | 0.8465 | 0.9602 | 0.9251 | 0.9751 | 0.9597 | 0.9997 | 3.4662 | 2.4597 |
| 10 | 0.5128 | 0.8172 | 0.6707 | 0.9705 | 0.7687 | 0.9934 | 9.8032 | 2.2849 |

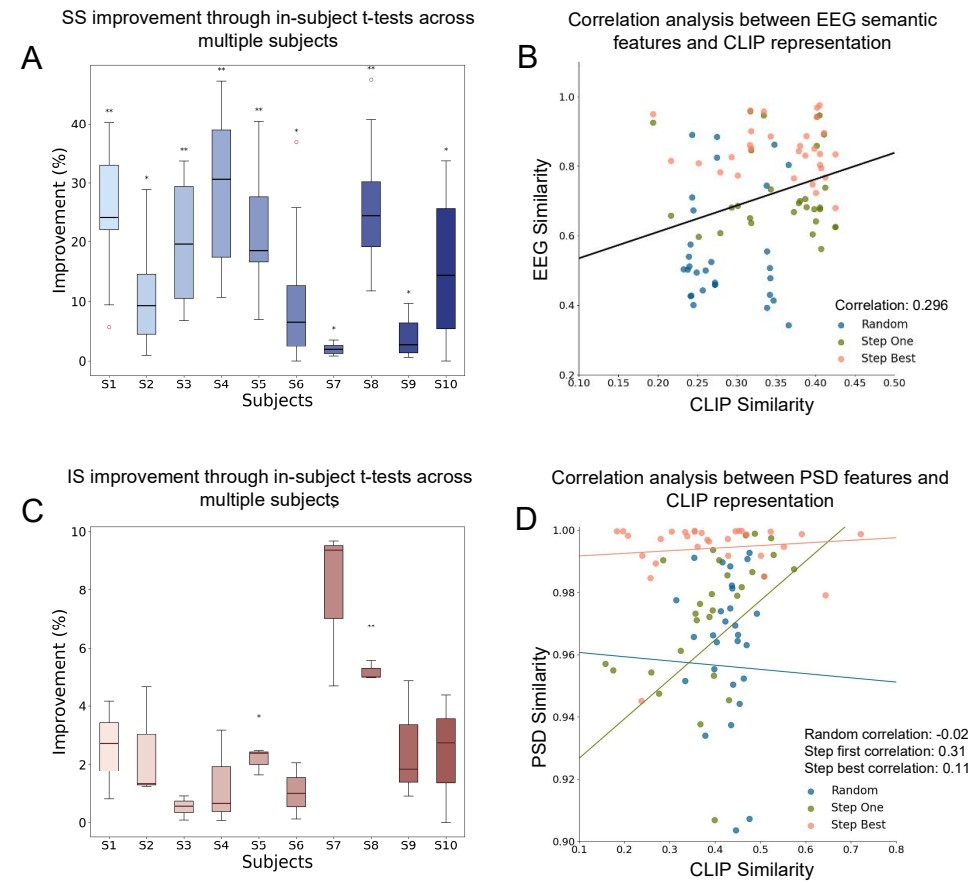

Figure A.3: **Improvement in similarity scores assessed via paired t-tests and correlation of similarity scores with targets across all subjects.** (A) Average EEG semantic representation scores (SS) for various target EEG semantic features. (B) The correlation of the similarity score with target between EEG semantic features across all subjects. (C) Average EEG response intensity scores (IS) for different target EEG PSD features. (D) The correlation of the similarity score with target between EEG PSD features across all subjects.

A.3  VALIDITY VERIFICATION OF SYNTHETIC EEG

To evaluate the performance of our EEG encoding models, we compare the synthetic EEG signals generated by two deep neural networks (DNNs)—AlexNet and CORnet-S—with real EEG data. Here's a step-by-step breakdown of how we processed and compared the data.

We selected 17 specific channels from the original 63-channel EEG dataset, focusing on those most relevant to visual processing. It ensured that we focused on neural regions most directly involved in responding to the visual stimuli. For each stimulus, we averaged the EEG signals across all trials, resulting in a representative dataset for each stimulus. This reduced the dimensionality of the data, making it easier to compare with synthetic data. We used a pretrained end-to-end encoding model to generate synthetic EEG signals based on the visual stimuli. The model captures the mapping between the visual input and the resulting EEG signals using deep neural networks. These synthetic signals represent the neural responses predicted by the model in response to the stimuli.

Table A.2: MSE Values for synthesized EEG

| Subject | Pretrained | | Random Init | | Average |
|---|---|---|---|---|---|
| | AlexNet | CORnet-S | AlexNet | CORnet-S | |
| Sub-01 | 0.1095 | 0.1126 | 0.1161 | 0.0994 | 0.1094 |
| Sub-02 | 0.0764 | 0.0788 | 0.0840 | 0.0994 | 0.0847 |
| Sub-03 | 0.0787 | 0.0806 | 0.0816 | 0.0910 | 0.0830 |
| Sub-04 | 0.0652 | 0.0664 | 0.0662 | 0.1011 | 0.0747 |
| Sub-05 | 0.0493 | 0.0515 | 0.0704 | 0.0975 | 0.0672 |
| Sub-06 | 0.0690 | 0.0719 | 0.0498 | 0.0966 | 0.0718 |
| Sub-07 | 0.1267 | 0.1300 | 0.0914 | 0.1312 | 0.1198 |
| Sub-08 | 0.0718 | 0.0727 | 0.1038 | 0.1165 | 0.0912 |
| Sub-09 | 0.0529 | 0.0563 | 0.0781 | 0.0756 | 0.0657 |
| Sub-10 | 0.1122 | 0.1151 | 0.0961 | 0.1149 | 0.1096 |
| Average | 0.0810 | 0.0832 | 0.0838 | 0.1023 | 0.0876 |

Table A.2 presents the mean squared error (MSE) between the synthetic EEG signals generated by AlexNet and CORnet-S, and the real EEG signals for 10 subjects. The MSE was computed for each individual test sample and then averaged across the entire test set. Lower MSE values indicate better alignment between the synthetic and real EEG signals.

From the comparison shown in the Figure A.4, the retrieval accuracy for S-S (both training and testing sets consist of generated signals) is significantly higher than other categories, including T-T (both training and testing sets consist of real signals), T-S (training set consists of real signals, testing set consists of generated signals), and S-T (training set consists of generated signals, testing set consists of real signals), under both AlexNet and CORnet-S models. This indicates:

**Advantages of generated signals** Supported by black-box ANN models (e.g., AlexNet and CORnet-S), generated signals perform significantly better in retrieval tasks compared to real signals. In particular, the highest retrieval accuracy for S-S demonstrates the consistency and model adaptability of generated signals in this retrieval task.

**Model adaptability**: Different ANN models (e.g., AlexNet and CORnet-S) show consistent superiority in the retrieval tasks for generated signals, indicating that generated signals are more easily captured and distinguished by black-box models.

In Figure A.5, we compute the variance across all samples and time points for each channel, providing a measure of the overall variability of the EEG signals in response to different visual stimuli and their temporal dynamics. This variance can help identify channels with the highest variability, which may be useful for selecting specific channels for further analysis or modulation.

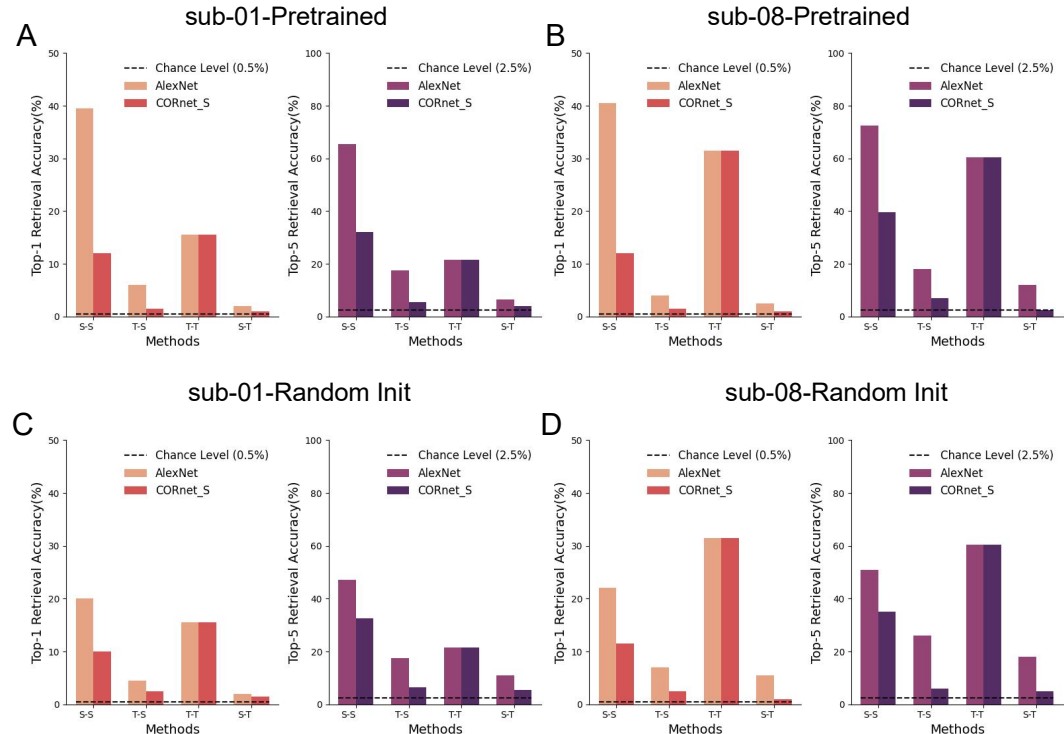

Figure A.4: Retrieval accuracy under different training and test datasets. Zero-shot retrieval performance of EEG data from different sources in Subject 1 and Subject 8 using ATM-S in different Settings. AlexNet and CORnet-S used in the first row were both pre-trained end-to-end models, and the second row was randomly initialized end-to-end.

In Figure A.6, we show the variance and standard deviation of the EEG signals computed across samples for each time point, and then averaged across channels. This analysis allows us to assess how signal variability evolves over time. By comparing the real EEG data with synthetic data generated by AlexNet and CORnet-S, we can evaluate how well each model captures the temporal variability present in the real EEG signals.

In Figure A.7, we compute the Pearson correlation coefficient between the averaged real EEG data and the synthetic data for each stimulus, measuring how well the synthetic data matches the real EEG on a per-sample basis. The histogram shows the distribution of correlation coefficients across all samples for both AlexNet and CORnet-S. A higher concentration of peaks near higher Pearson coefficients indicates better alignment between the synthetic data and the real EEG, reflecting superior model performance.

In Figure A.8, for each time point, we compute the Pearson correlation between the real EEG signal and the synthetic signals. This analysis enables us to visualize how well each model replicates the temporal structure of real neural responses to visual stimuli. Shaded regions in the plot represent the standard deviation across samples, showing the variability in model performance over time. The results provide a detailed view of how each model performs at different time points, highlighting which model more accurately captures the temporal dynamics of EEG signals.stimuli.

From the above analysis, we observe that both AlexNet and CORnet-S perform well, showing comparable results in terms of MSE, spatial (channel-wise) variability, and temporal (time-resolved) variability. The Pearson correlation analysis further confirms that both models synthesize EEG signals that align well with real data, with subtle differences in performance between them. These findings highlight the robustness of our EEG encoding models, demonstrating their ability that not only mimic the structural features of real EEG data but also capture the realistic variability seen in neural responses to visual stimuli. This suggests that our models are effective in approximating the neural representations underlying visual processing.

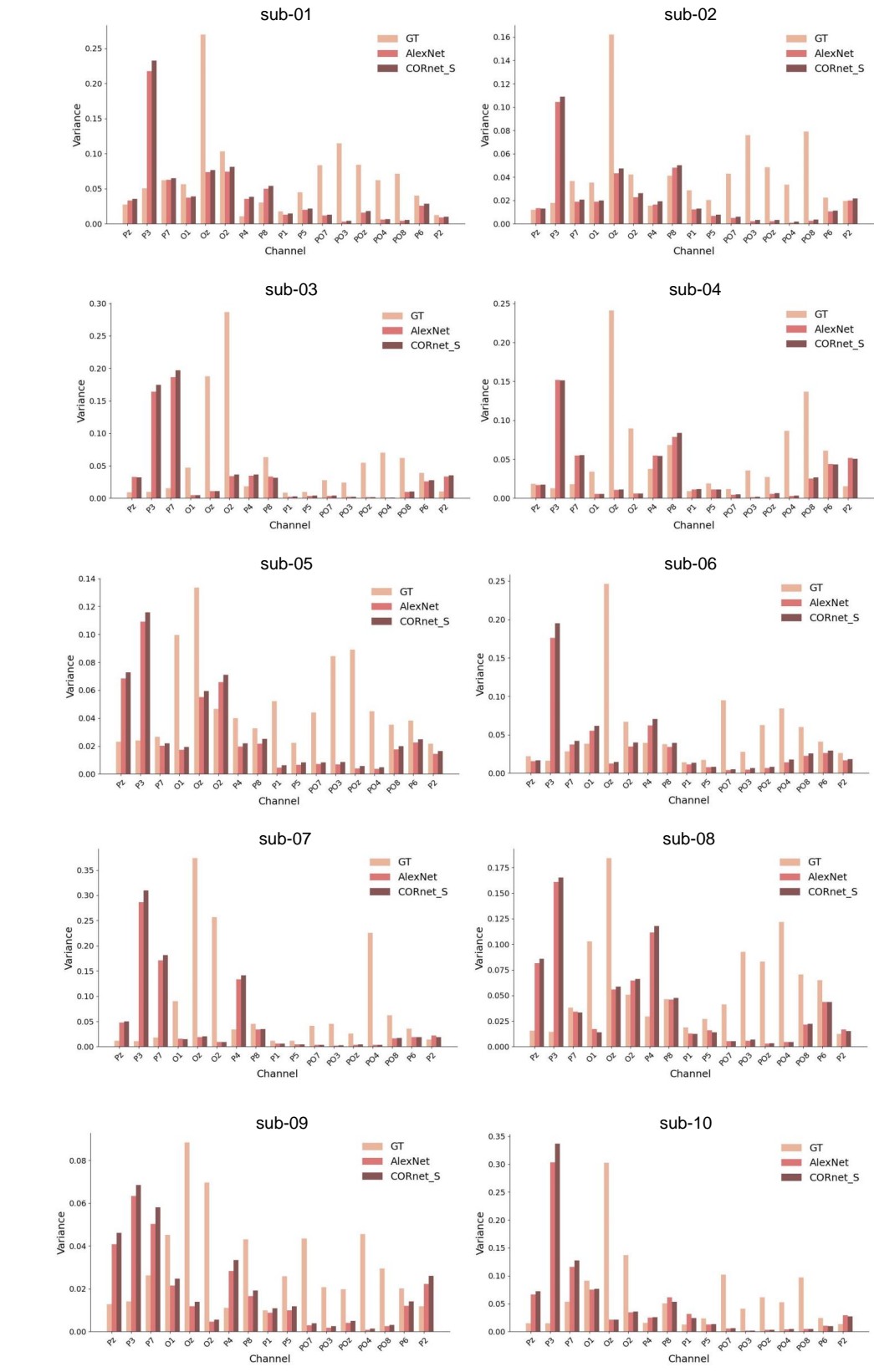

Figure A.5: Variance across different channels for different visual stimulus and temporal dynamics

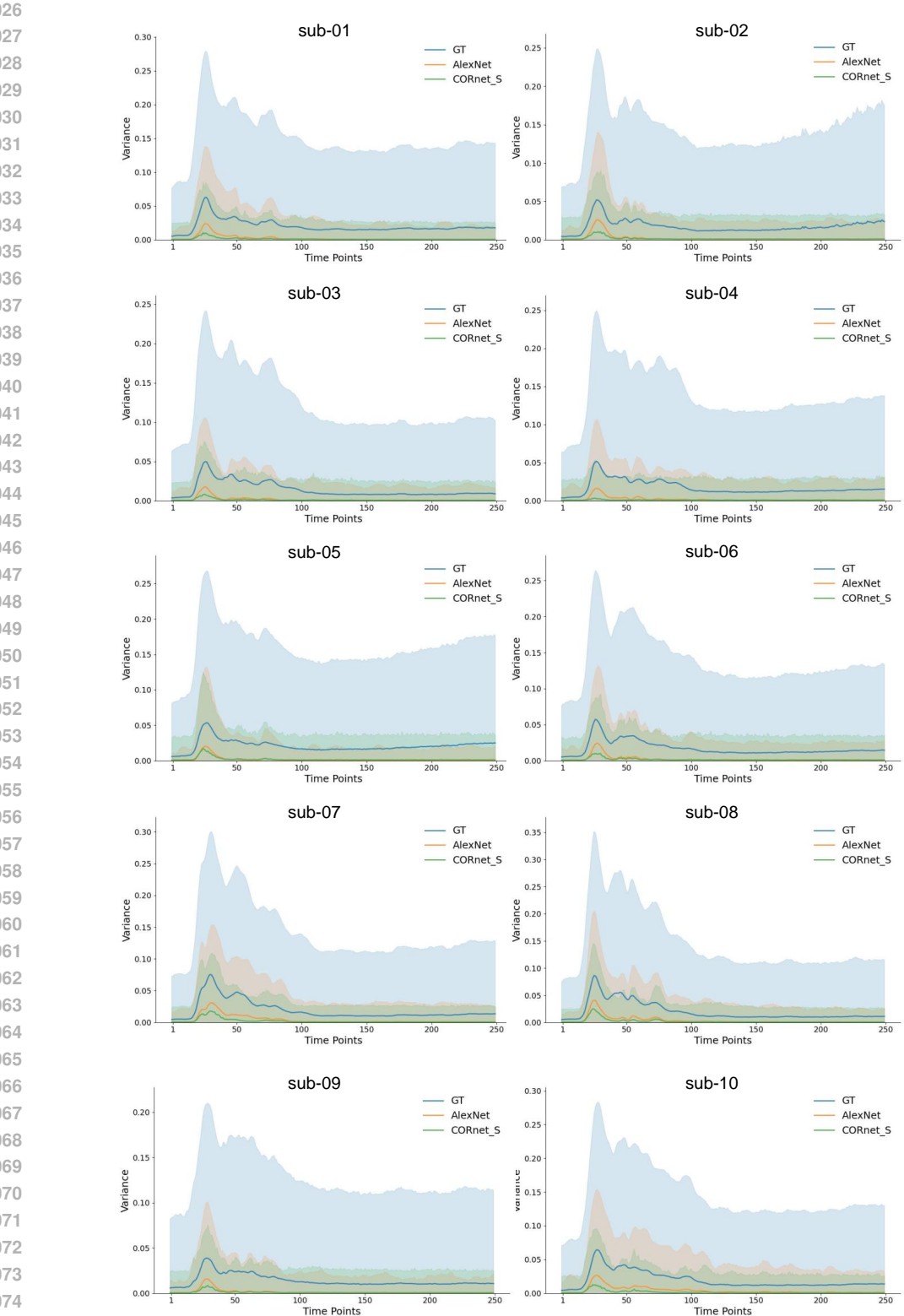

Figure A.6: Variance across different time points for different visual stimuli and channels.

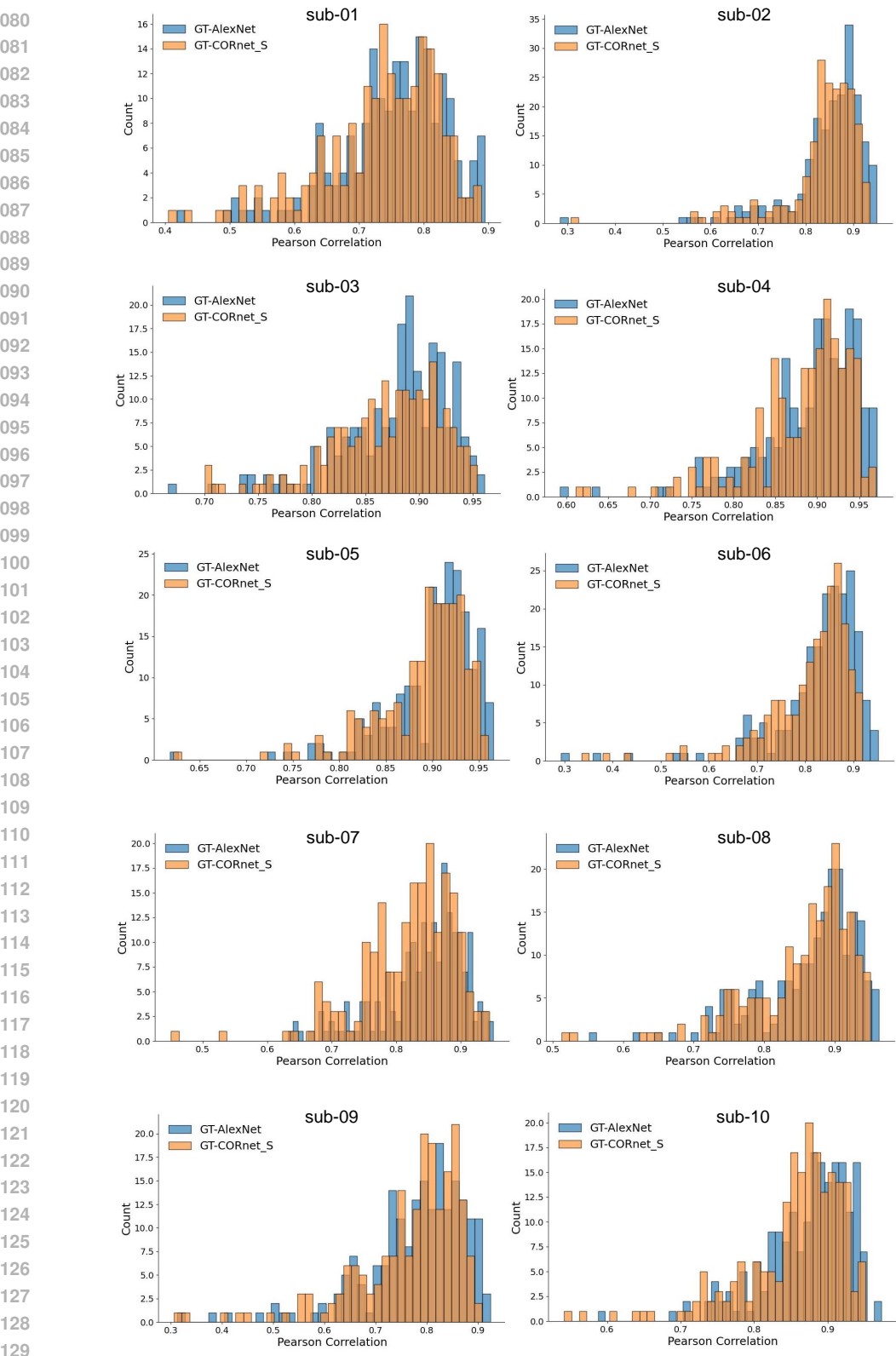

Figure A.7: Distribution of Pearson correlation coefficients across all sample pairs.

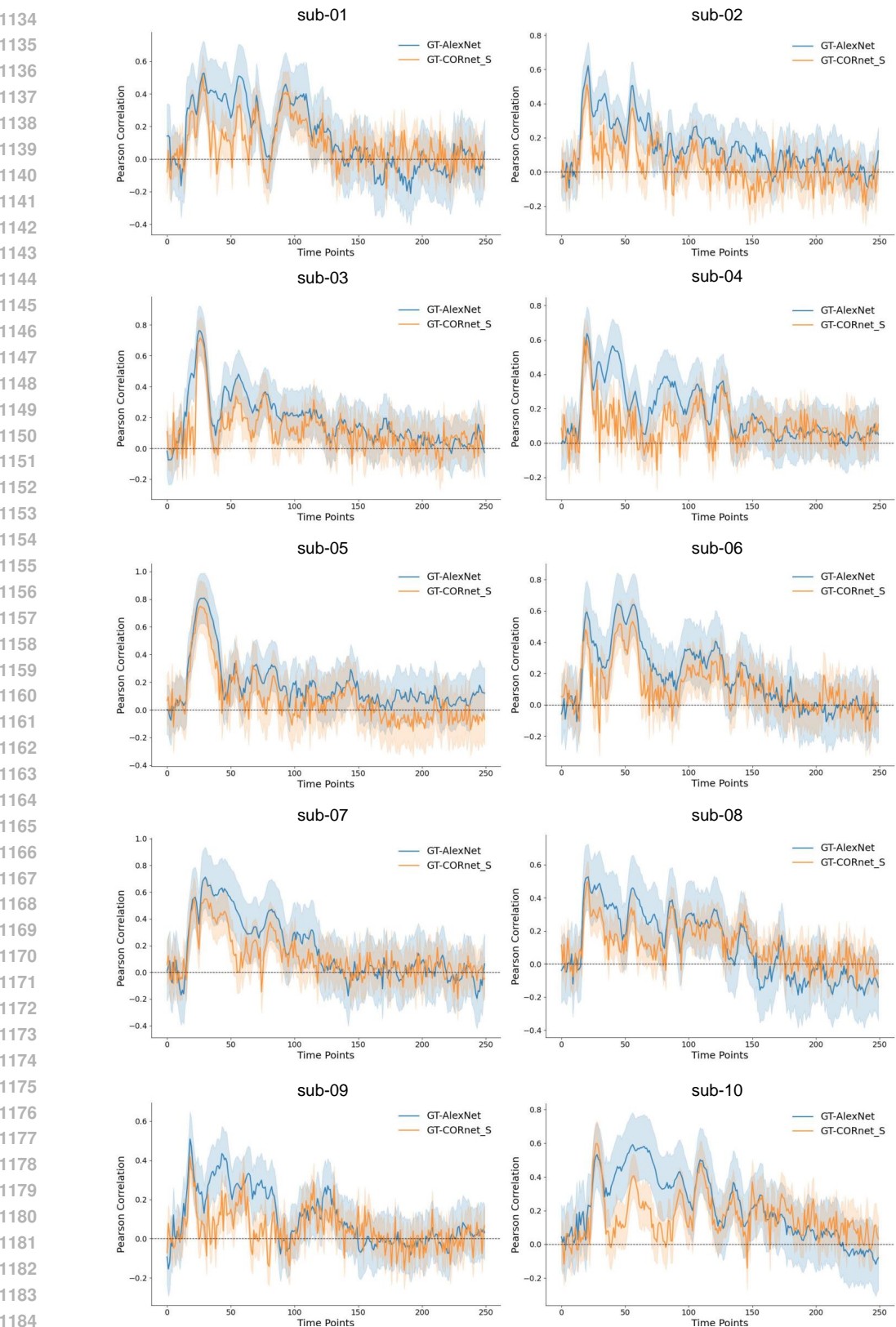

Figure A.8: Time-resolved Pearson correlation between ground truth EEG signals and synthetic EEG signals predicted by two neural network models (AlexNet and CORnet-S).

## A.4 ADDITIONAL RETRIEVAL EXAMPLES OF SEMANTIC REPRESENTATION

### A.4.1 MORE EXAMPLES OF RETRIEVAL

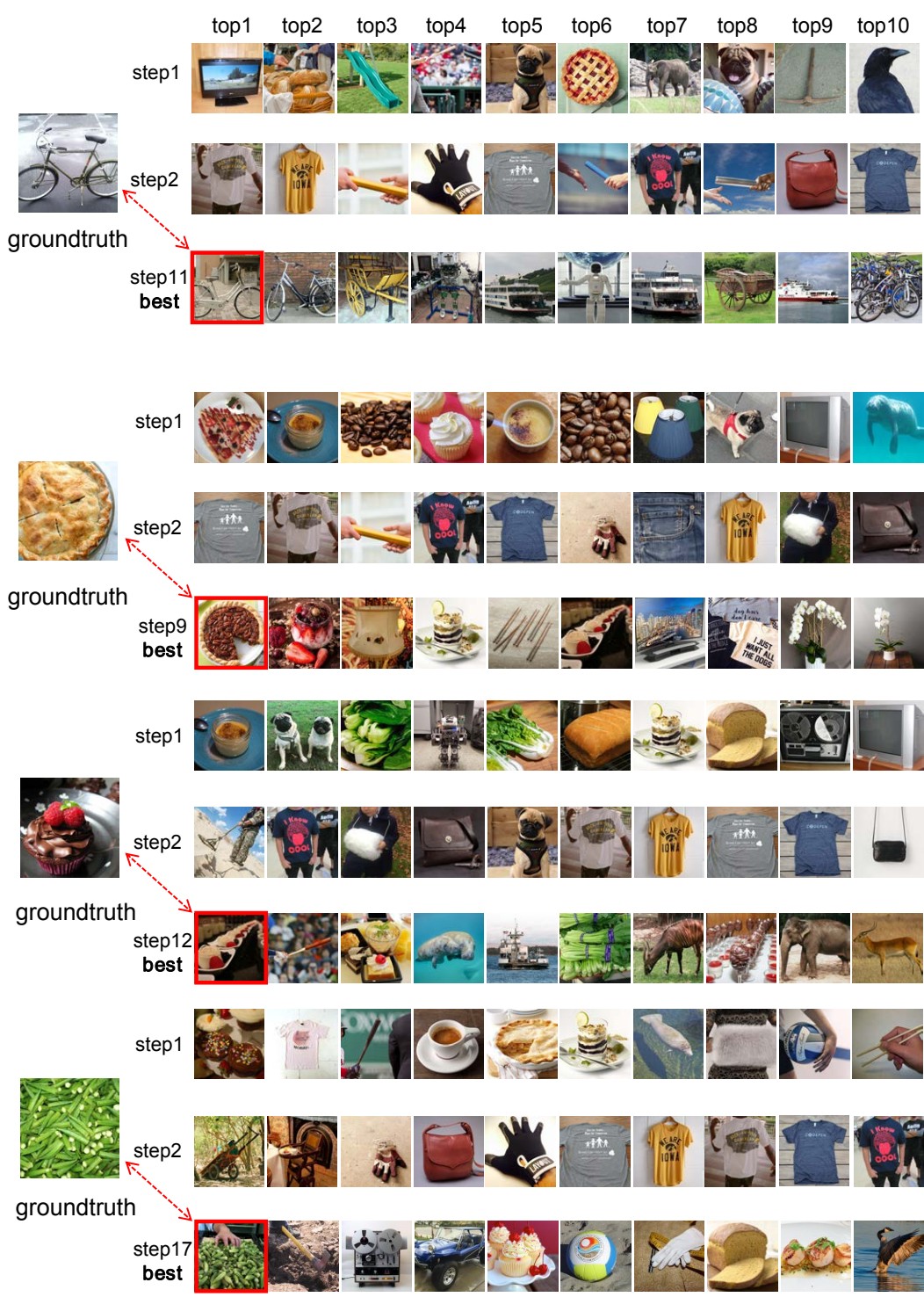

Figure A.9: **Some retrieval examples of Subject 8, 4, 4, and 1.** By setting different targets, we present examples where the stimulus retrieved at the end of the iterative optimization process increasingly approximates the true category.

### A.4.2 SOME FAILURE EXAMPLES OF RETRIEVAL

Figure A.10: **Some retrieval failure examples of Subject 8.** By setting different targets, we show examples where the stimulus retrieved at the end of the iteration is far from the true category. In these examples, the final retrieved stimulus exhibits varying degrees of similarity to the target image.

A.5 ADDITIONAL CONTROLLABLE GENERATION EXAMPLES OF PSD FEATURE

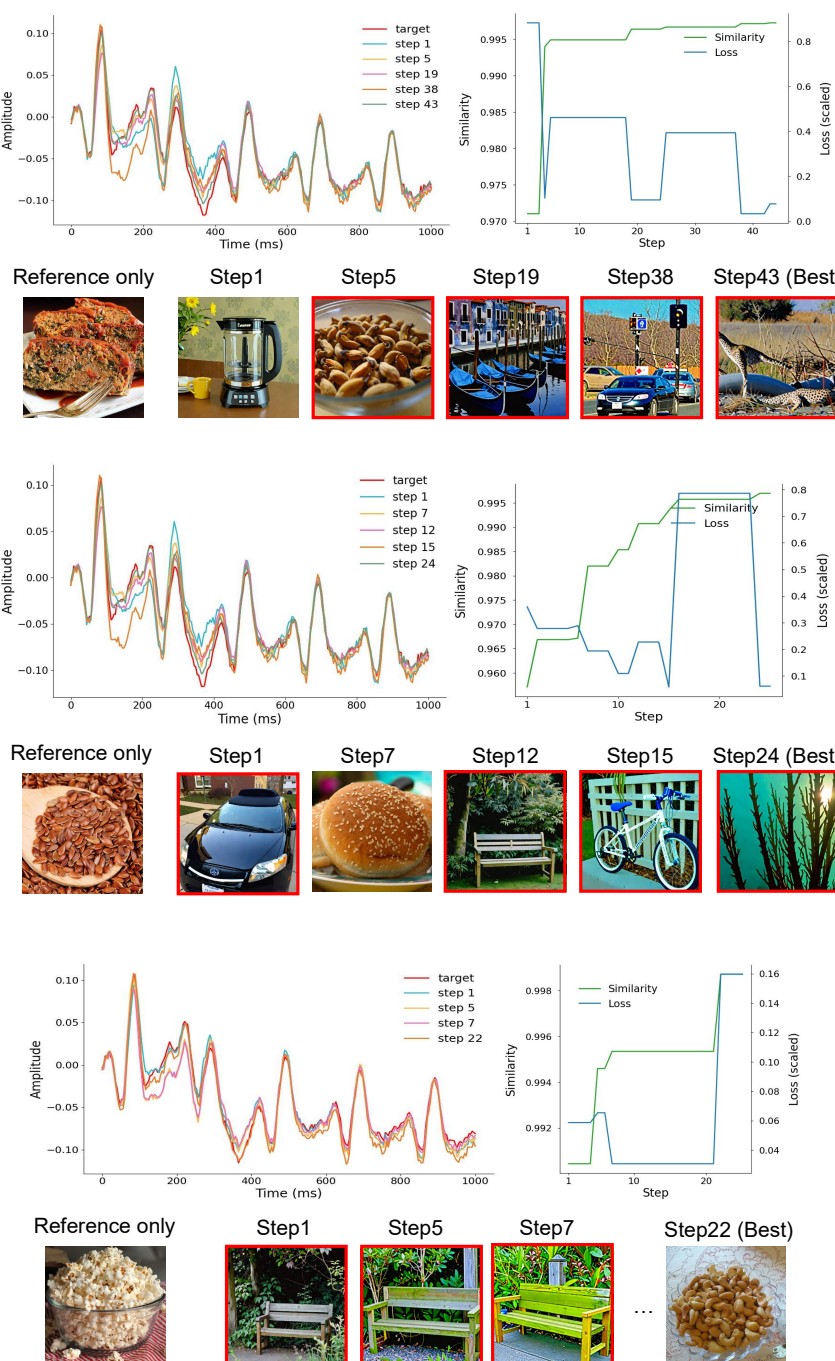

Figure A.11: **Illustration of the closed-loop iterative process for Subject 1.** Three distinct visual targets were presented, each based on a specific similarity measure (details in Target Features of EEG, Section 4.1), with new visual stimuli iteratively generated for each target. The left panel illustrates the time-domain evolution of neural responses across iterations. The right panel depicts the changes in similarity (green curve) and loss (blue curve, scaled) between the current stage features and the target features.

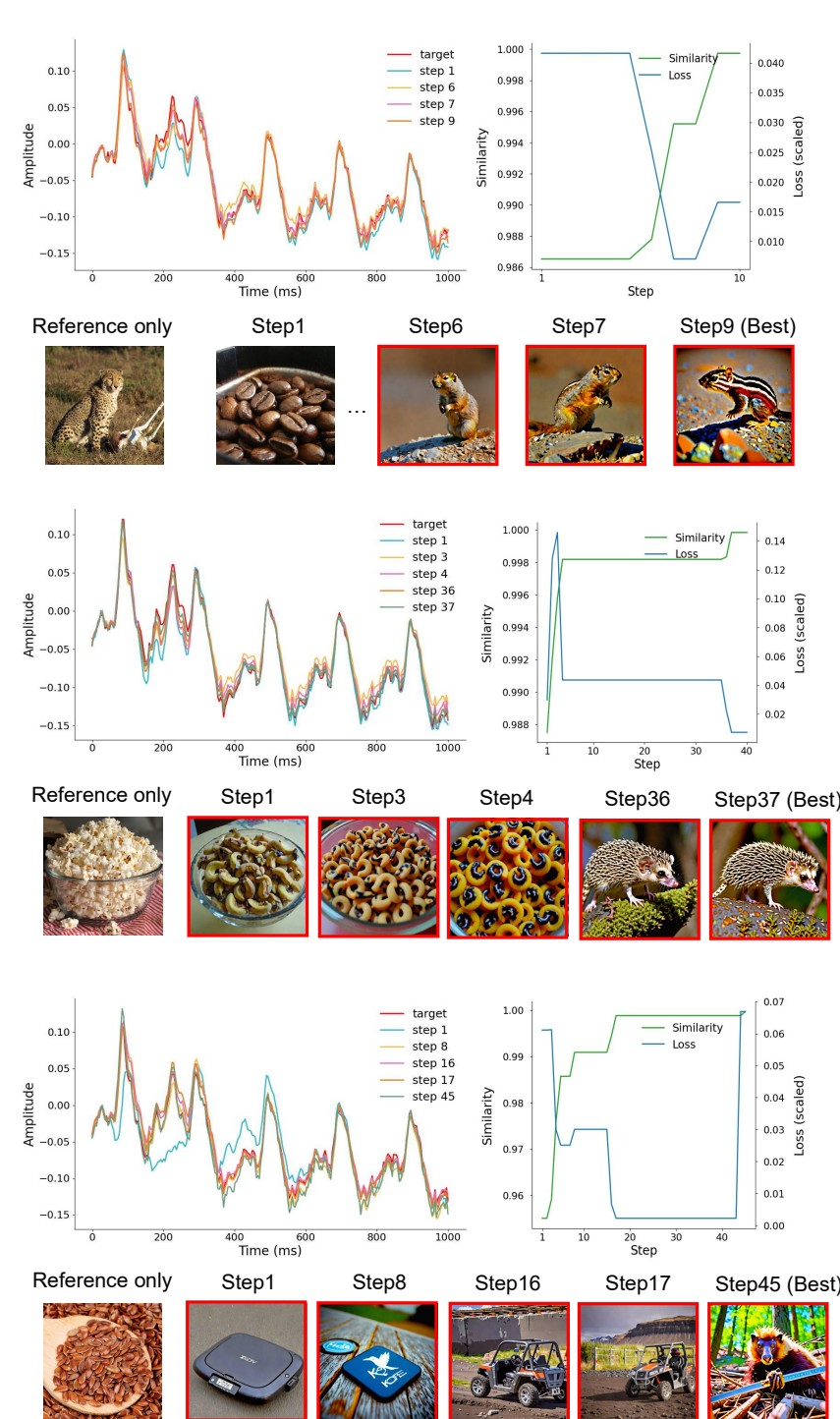

Figure A.12: **Illustration of the closed-loop iterative process for Subject 2.** Three distinct visual targets were presented, each based on a specific similarity measure (details in Target Features of EEG, Section 4.1), with new visual stimuli iteratively generated for each target. The left panel illustrates the time-domain evolution of neural responses across iterations. The right panel depicts the changes in similarity (green curve) and loss (blue curve, scaled) between the current stage features and the target features.

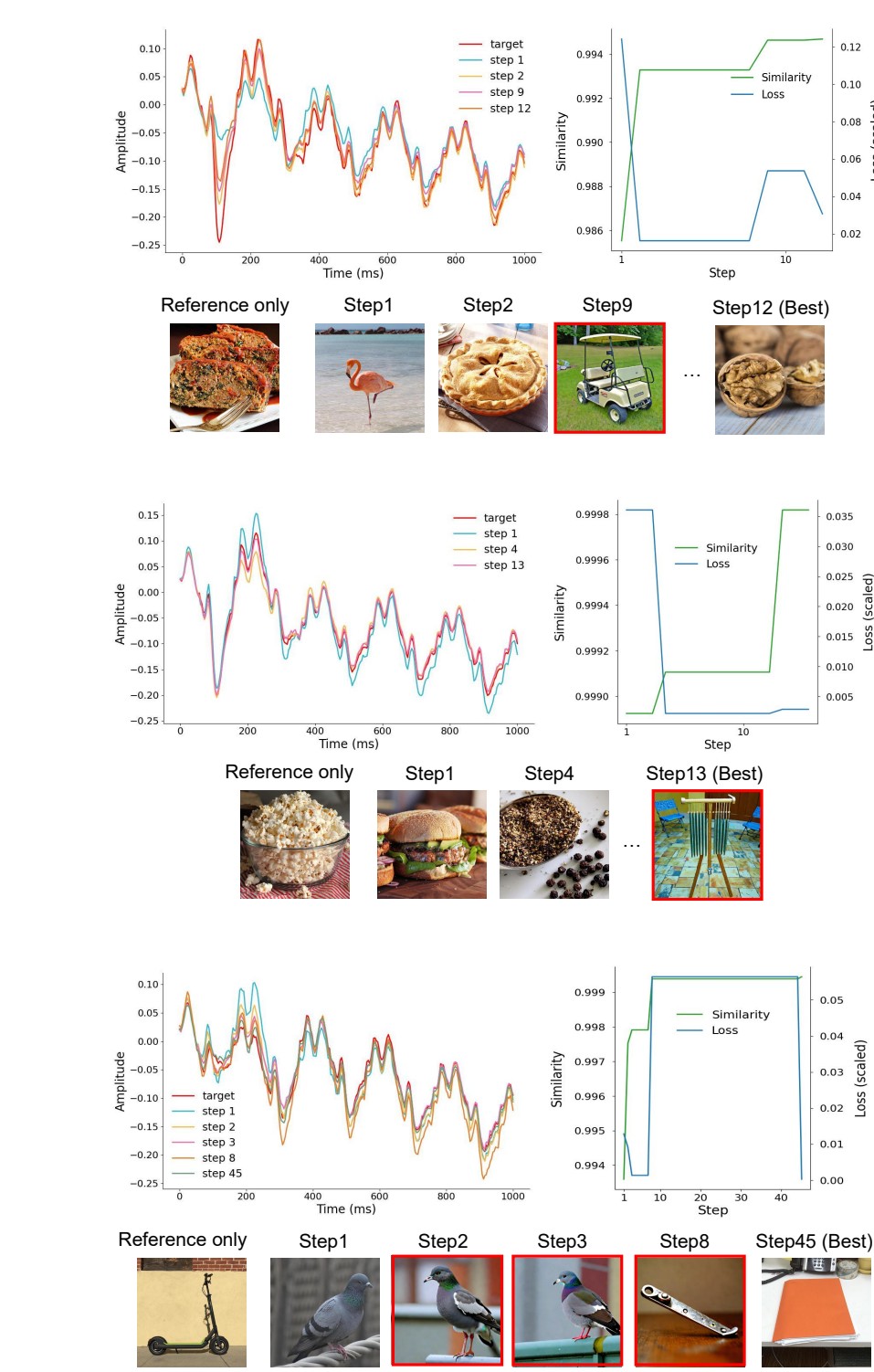

Figure A.13: **Illustration of the closed-loop iterative process for Subject 3.** Three distinct visual targets were presented, each based on a specific similarity measure (details in Target Features of EEG, Section 4.1), with new visual stimuli iteratively generated for each target. The left panel illustrates the time-domain evolution of neural responses across iterations. The right panel depicts the changes in similarity (green curve) and loss (blue curve, scaled) between the current stage features and the target features.

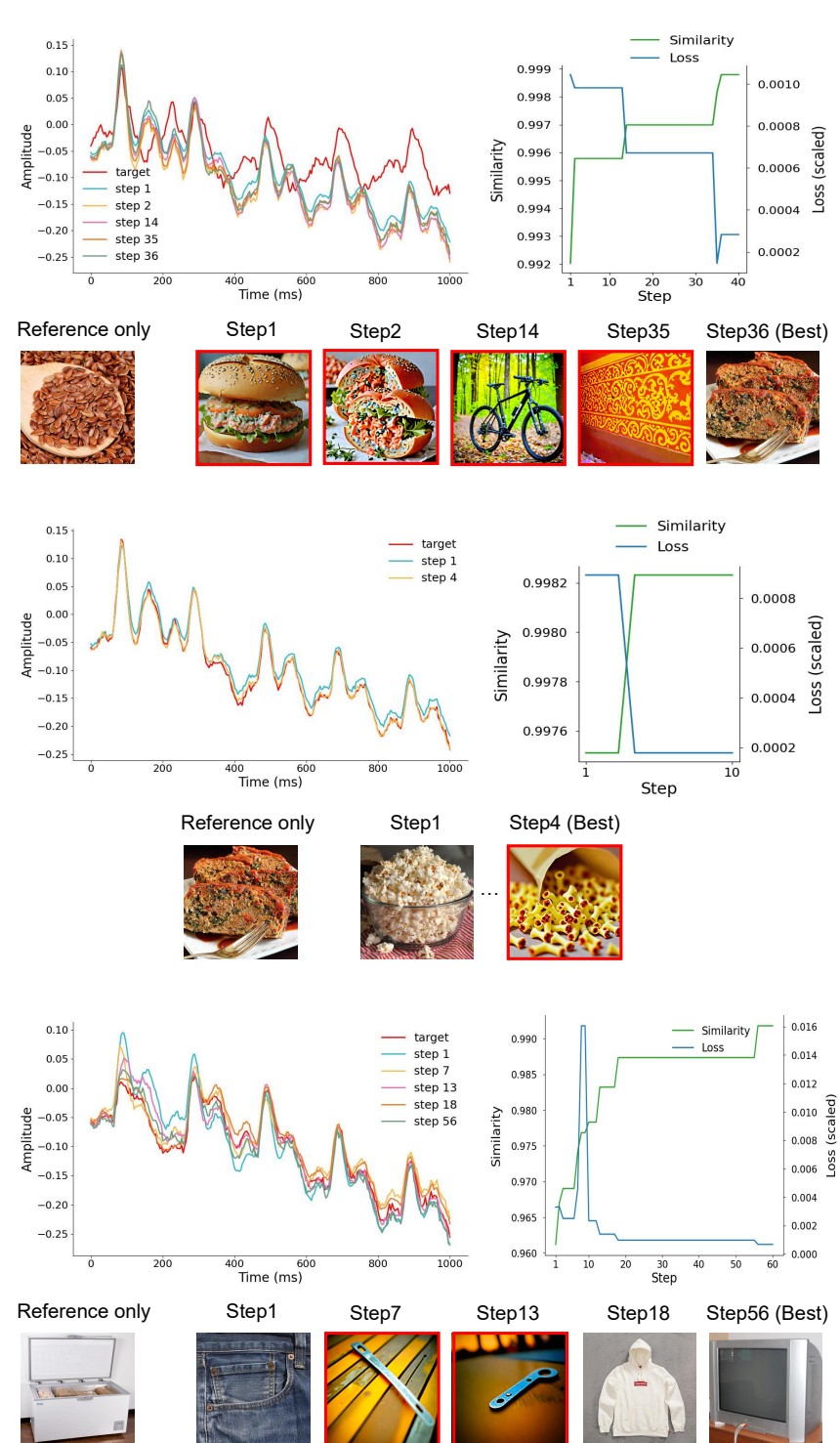

Figure A.14: **Illustration of the closed-loop iterative process for Subject 4.** Three distinct visual targets were presented, each based on a specific similarity measure (details in Target Features of EEG, Section 4.1), with new visual stimuli iteratively generated for each target. The left panel illustrates the time-domain evolution of neural responses across iterations. The right panel depicts the changes in similarity (green curve) and loss (blue curve, scaled) between the current stage features and the target features.

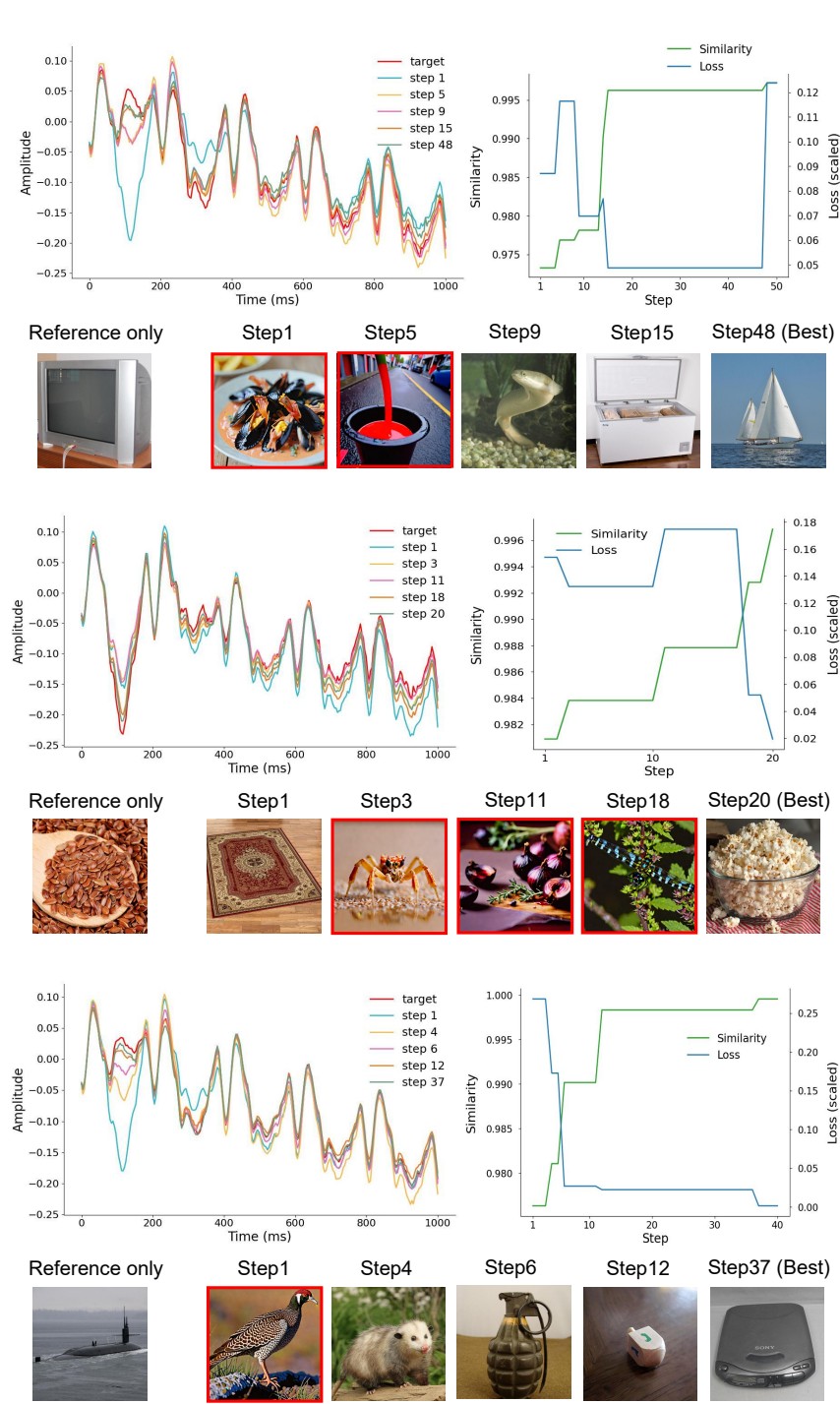

Figure A.15: **Illustration of the closed-loop iterative process for Subject 5.** Three distinct visual targets were presented, each based on a specific similarity measure (details in Target Features of EEG, Section 4.1), with new visual stimuli iteratively generated for each target. The left panel illustrates the time-domain evolution of neural responses across iterations. The right panel depicts the changes in similarity (green curve) and loss (blue curve, scaled) between the current stage features and the target features.

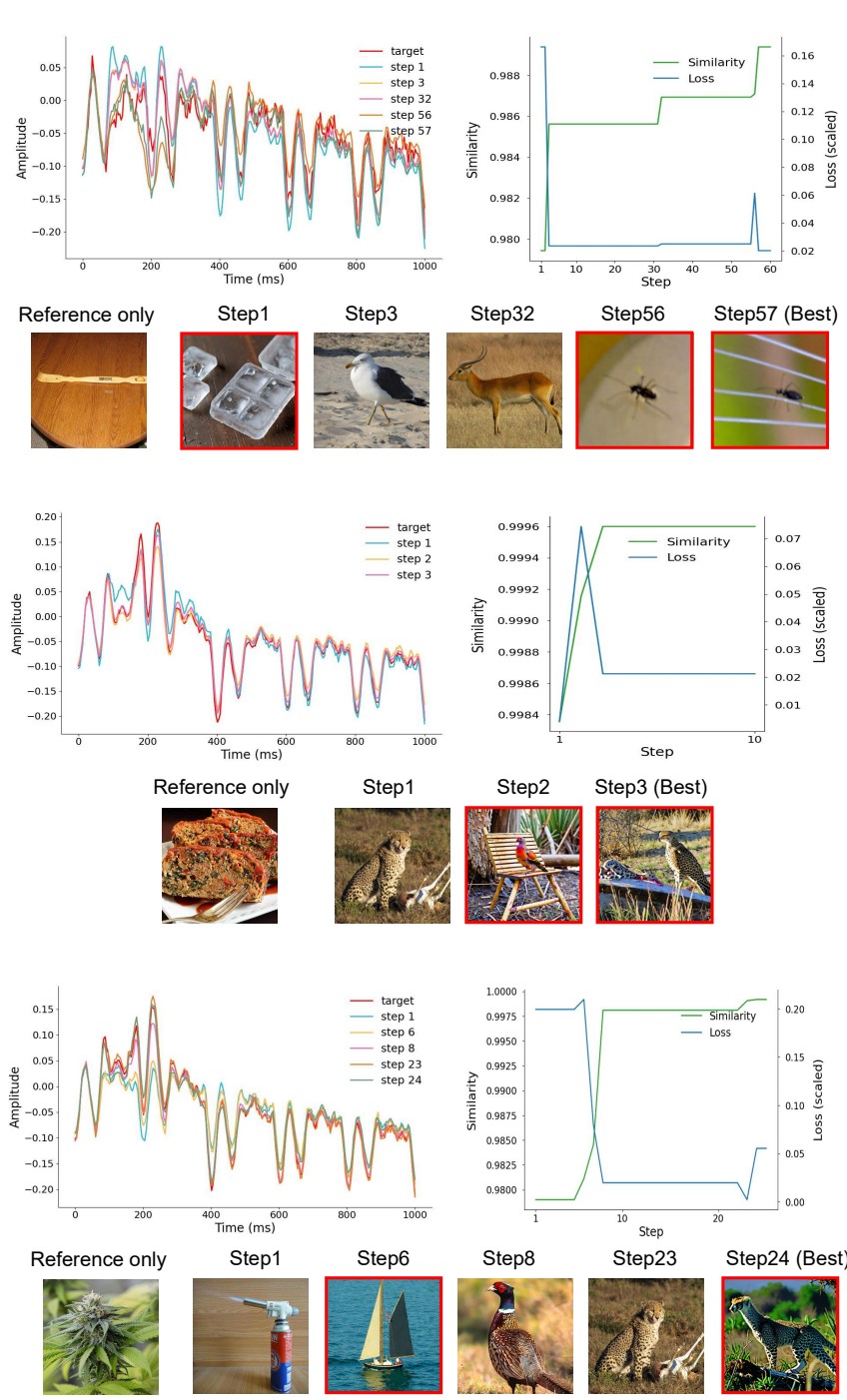

Figure A.16: **Illustration of the closed-loop iterative process for Subject 6.** Three distinct visual targets were presented, each based on a specific similarity measure (details in Target Features of EEG, Section 4.1), with new visual stimuli iteratively generated for each target. The left panel illustrates the time-domain evolution of neural responses across iterations. The right panel depicts the changes in similarity (green curve) and loss (blue curve, scaled) between the current stage features and the target features.

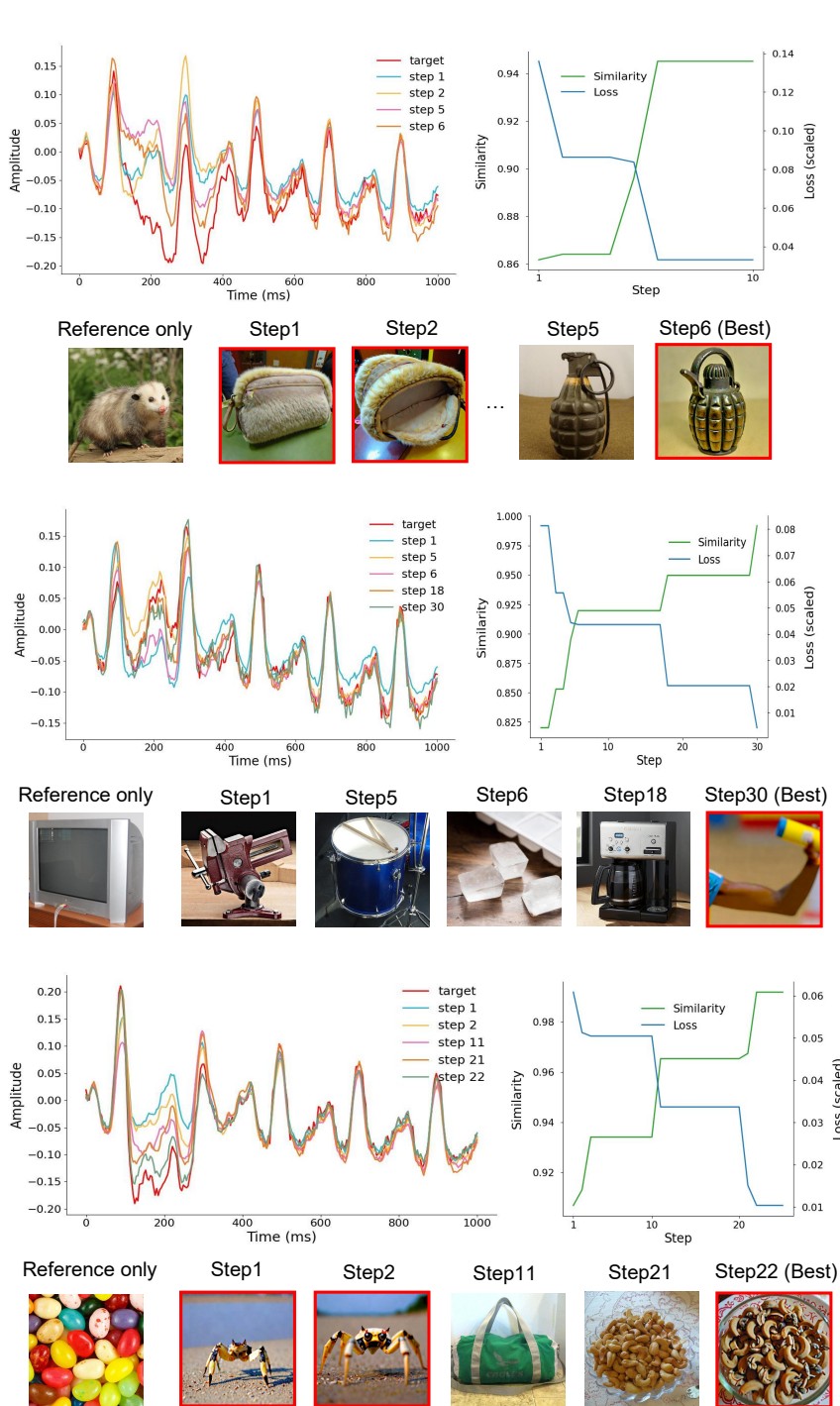

Figure A.17: **Illustration of the closed-loop iterative process for Subject 7.** Three distinct visual targets were presented, each based on a specific similarity measure (details in Target Features of EEG, Section 4.1), with new visual stimuli iteratively generated for each target. The left panel illustrates the time-domain evolution of neural responses across iterations. The right panel depicts the changes in similarity (green curve) and loss (blue curve, scaled) between the current stage features and the target features.

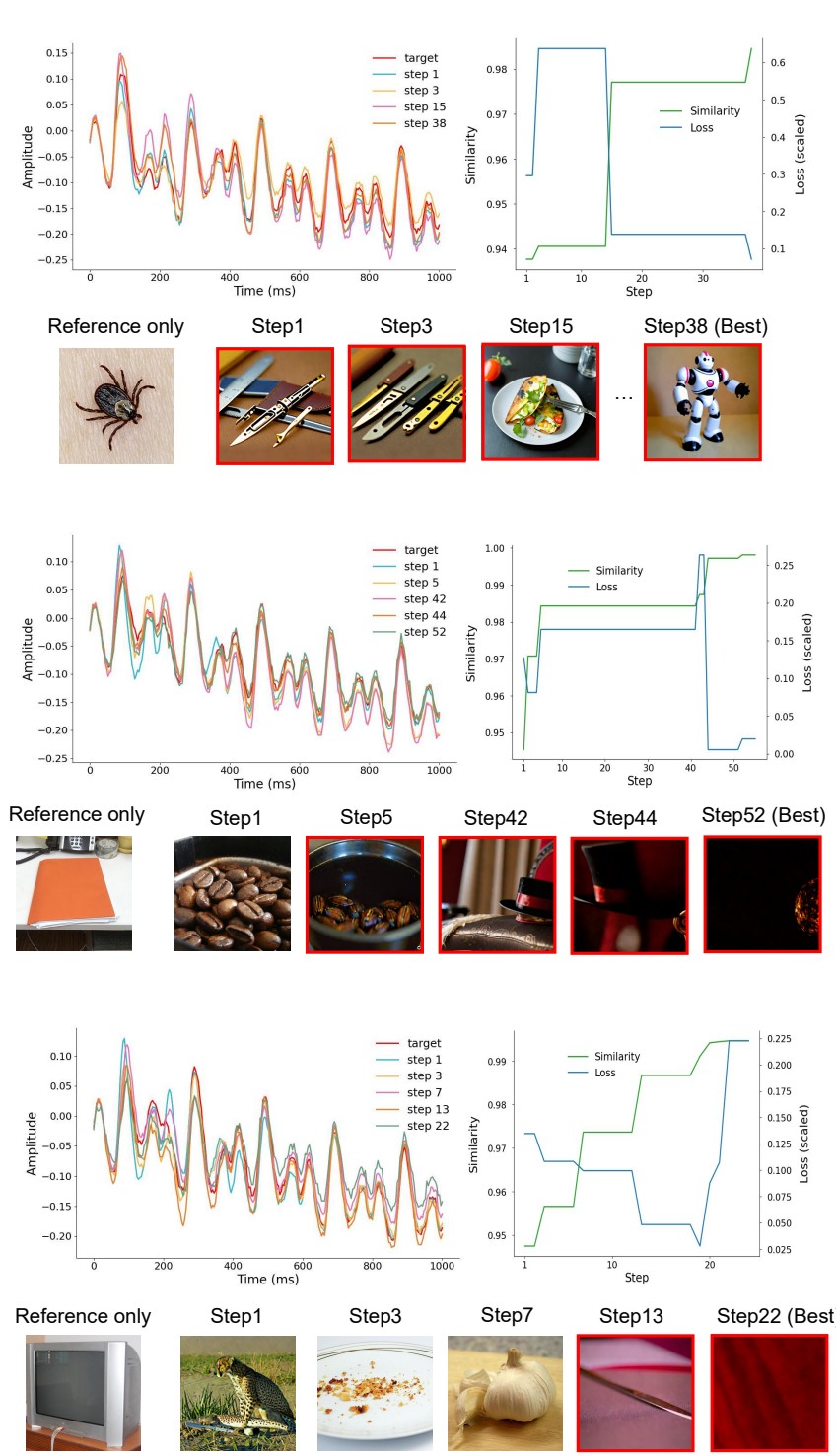

Figure A.18: **Illustration of the closed-loop iterative process for Subject 8.** Three distinct visual targets were presented, each based on a specific similarity measure (details in Target Features of EEG, Section 4.1), with new visual stimuli iteratively generated for each target. The left panel illustrates the time-domain evolution of neural responses across iterations. The right panel depicts the changes in similarity (green curve) and loss (blue curve, scaled) between the current stage features and the target features.

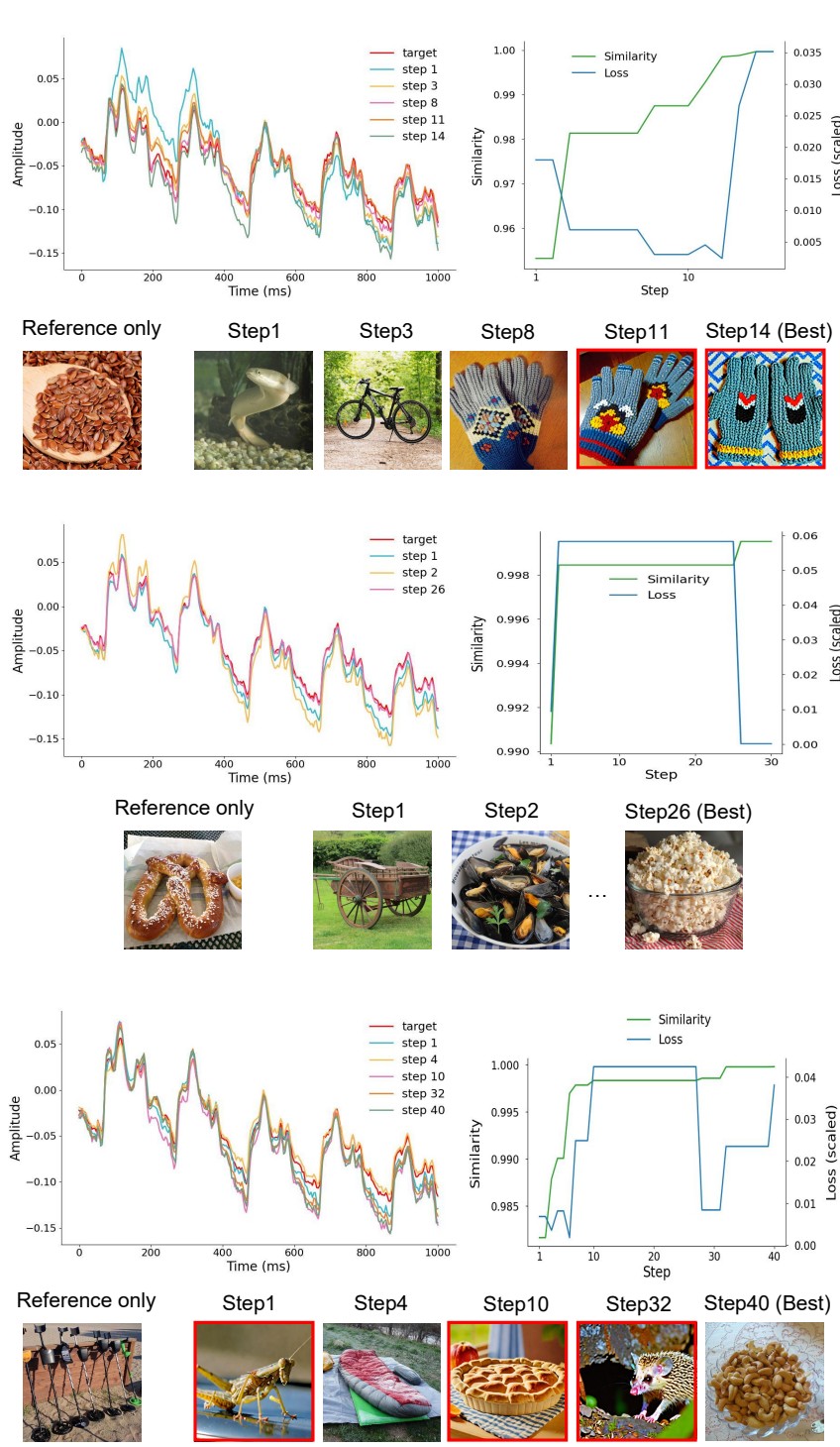

Figure A.19: **Illustration of the closed-loop iterative process for Subject 9.** Three distinct visual targets were presented, each based on a specific similarity measure (details in Target Features of EEG, Section 4.1), with new visual stimuli iteratively generated for each target. The left panel illustrates the time-domain evolution of neural responses across iterations. The right panel depicts the changes in similarity (green curve) and loss (blue curve, scaled) between the current stage features and the target features.

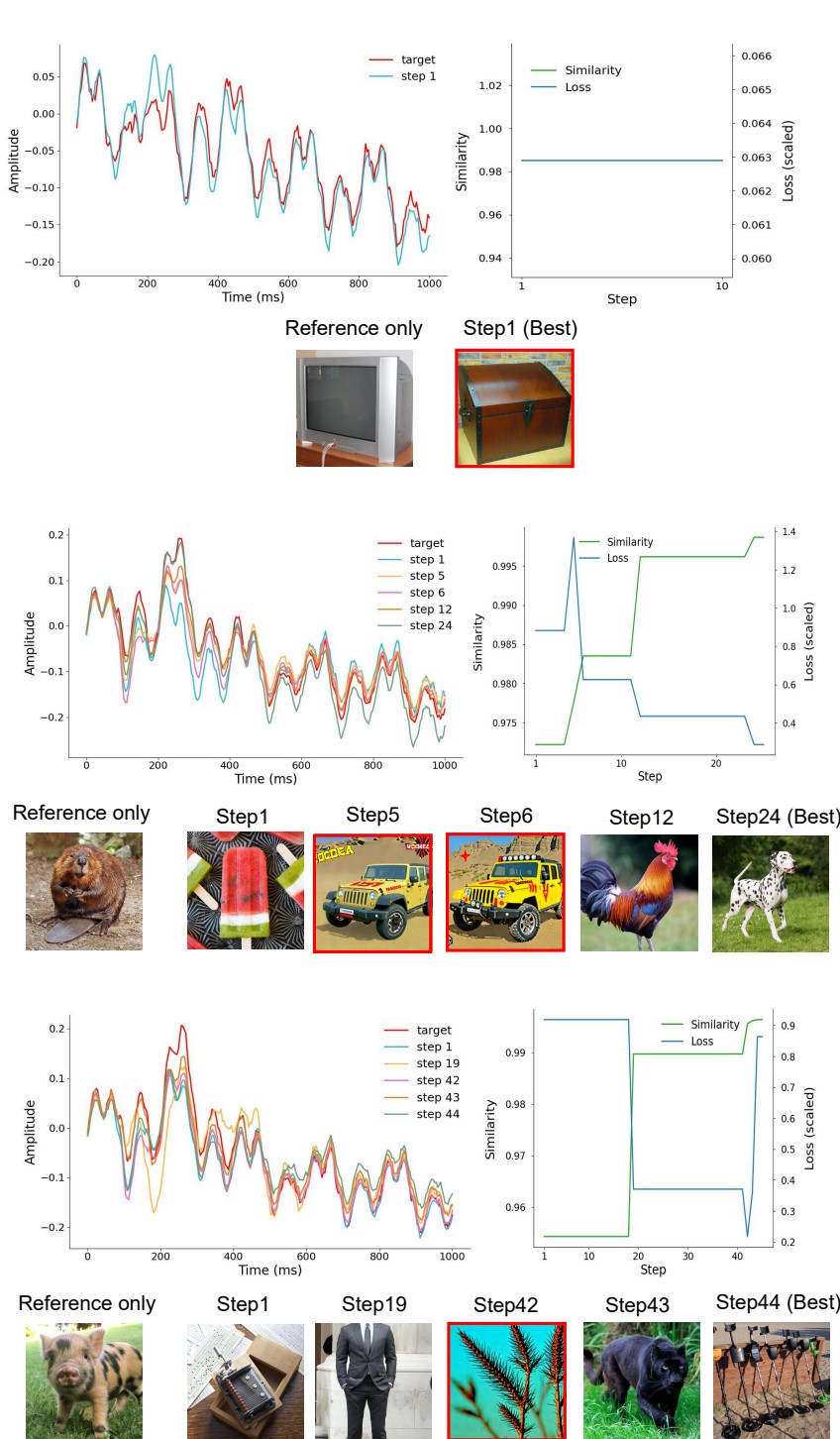

Figure A.20: **Illustration of the closed-loop iterative process for Subject 10.** Three distinct visual targets were presented, each based on a specific similarity measure (details in Target Features of EEG, Section 4.1), with new visual stimuli iteratively generated for each target. The left panel illustrates the time-domain evolution of neural responses across iterations. The right panel depicts the changes in similarity (green curve) and loss (blue curve, scaled) between the current stage features and the target features.

