# OpenReview forum: "A closed-loop EEG-based visual stimulation framework from controllable generation"
_ICLR.cc/2025/Conference — Submitted to ICLR 2025_

### Official Review · Reviewer_4DNY · 2024-11-01

**Soundness:** 3
**Presentation:** 1
**Contribution:** 2
**Rating:** 3
**Confidence:** 4

**Summary:**

This paper devised a closed-loop framework to find the visual stimuli that can elicit specific neural activities. The authors models the whole process as an MDP, and proposed to use the interactive search (mind matching) and heuristic search (genetic algorithm) to solve the problem. While claimed general, the authors specify the framework to  train the EEG encoding model to generate the synthesized EEG response and test it offline on the THINGS-EEG2 dataset. Visualized results demonstrate the possibility of the whole framework to find the appropriate visual stimuli in the search space. The authors also mentioned its possible impact and insights.

**Strengths:**

1. The whole framework is novel and interesting. It addresses the challenge to find the corresponding stimuli that can evoke a specific brain signal pattern. The framework may have the potential to be applied to a more realistic scenario.
2. The paper proposed two different settings for finding the visual stimuli: retrieval and generation, and provided corresponding solutions for them.
3. The overall findings may provide interesting neuroscience intuitions and may ignite further contributions.

**Weaknesses:**

1. One of the main claims by the authors is the adaptation of the whole close-loop framework. While the authors claim it can be simply replaced by recording EEG data from human participants, there are actually no more concrete demonstrations on how. For example, what is the "specific neural activity in the brain" in this paper and in a possible real scenario? What's the difference? And how difficult is it and how much effort will it take to apply the framework to the real world? It's always easy to just claim a methodology "generalizable", but without more justification that doesn't actually help strengthen the contribution of the paper.
2. Based on 1, I feel it is not sufficiently demonstrated in the paper what role the EEG plays in the whole framework. As far as I can understand from the current paper, it seems to be related to the reward $R$ in the MDP design, because it should provide signal based on the desired neural activities. However, we know neither how the reward is exactly calculated nor what kinds of the neural signal the authors are caring about (e.g., a specific frequency bank? a specific shape of waveforms? a specific activation from some brain area?).
3. Besides the methodology, it's also not clear how the different part of this framework performs and contribute to the final result from the experimental aspect. While in the result section, we can see that the framework can yield promising visual stimuli result, it lacks either quantitative experiments and comparison between selection of algorithms, or a more detailed explanations on the presented ones. (See questions.) Therefore, it's unclear for me what the exact performance of the whole framework and individual parts compared to other solutions.
4. Overall, the presentation of this paper is unsatisfying (and that's probably why I have the concerns in 2 and 3). On the one hand, the author is presenting more well-known details in the main content but didn't make their own claims clear. For example, the algorithm 1 and algorithm 2 is a direct adaptation from previous work. Instead of using space to present them, I wish to see more on how the MDP is constructed. On the other hand, mixing citations with sentences (please use \citep instead \cite) and a few typos (in line 222, algorithm 1, the bracket is not matched) give me the feeling that the paper is not yet ready to be published.

**Questions:**

1. What kind of the neural activity are you concerning in your experiment? How will you verify whether the activity is properly stimulated by your visual stimuli?
2. If the answer to the previous question is via the EEG encoder, then how can the encoder capture your concerned neural activity? How does encoder perform? How will the selection of the encoder influence the result?
3. What is the reward in the MDP?
4. For Figure 3.B, why do you choose subject 8 for demonstration? It seems the confidence interval is large. I wonder whether the similarity increase can pass the significance test.
5. How to interpret the spectrograms in Figure 4.C? I can't see the difference or some trends from the figure.
6. How is Figure 4.D obtained? Why does the "random" also look so good?

---

> ### Author Response · Authors · 2024-11-21
> **Rebuttal by authors**
>
> **Q1: "What is the "specific neural activity in the brain" in this paper and in a possible real scenario? What's the difference? And how difficult is it and how much effort will it take to apply the framework to the real world? "**
>
> A1: We used the EEG data of each subject's training set to train the encoding model, and synthesized its EEG signal on the test set to simply replace the EEG data of human subjects. Because the real and specific neural activities of the subjects are complex and it is difficult to predict their dynamic characteristics for a long time, we compressed the specific neural activities into neural representations and approximated their neural activities by approximating the target neural representations.
>
> At present, the difficulty in applying this framework to the real world for online closed-loop control is that the amount of data from multiple rounds of feedback from each subject is scarce, and the quality of the collected signals is poor due to the influence of factors such as the environment and equipment. These factors lead to a slow convergence of the final algorithm. In addition, methods to improve the overall performance of the framework include finding suitable neural representations or using more advanced encoding models.
>
> **Q2: "...we know neither how the reward is exactly calculated nor what kinds of the neural signal the authors are caring about."**
>
> A2: Thank you for your careful consideration of this question. Our original manuscript missed some important details. We have revised the manuscript comprehensively. The specific revisions can be found in Global Response Q3. Unlike those works that directly control human behavior by generating images (Wei et al.), we introduce EEG as an observation of brain activity and control neural activity by controlling different types of neural representations of EEG. In Section 3.1, the reward R in the MDP design is the similarity score of the EEG features generated by the stimulus image of the current state and the target image. We introduce two different cases in Sections 3.3 and 3.4: EEG semantic representation and EEG channel energy representation, respectively. EEG semantic representation measures the information in the EEG corresponding to the semantics of the image category, which is obtained by pre-trained EEG Encoder aligned with CLIP. EEG channel energy representation corresponds to the PSD feature of a certain channel, reflecting the activation degree of the brain area, and is obtained by direct calculation.
>
> **Q3: "...It lacks either quantitative experiments and comparison between selection of algorithms, or a more detailed explanations on the presented ones. ...it's unclear for me what the exact performance of the whole framework and individual parts compared to other solutions."**
>
> A3: Thank you for pointing out this issue. We have revised the manuscript to clarify more technical details. In addition, we have added quantitative experiments in the Appendix A.2 and A.3, including the comparison of different encoding models, the verification of the effectiveness of the encoder and the quantitative results of all subjects. Our comprehensive revisions can be found in the updated PDF.
>
> **Q4: "I wish to see more on how the MDP is constructed. On the other hand, mixing citations with sentences (please use \citep instead \cite) and a few typos (in line 222, algorithm 1, the bracket is not matched) ..."**
>
> A4: Thank you for your suggestion, we have revised the manuscript and reported it in Global Response Q4. Please refer to our latest version PDF for more details.
>
> **Q5: "What kind of the neural activity are you concerning in your experiment? How will you verify whether the activity is properly stimulated by your visual stimuli?"**
>
> A5: Our framework attempts to summarize different neural activities into electrophysiological features, and use this type of features to guide the design of visual stimulation. Based on this, we proposed two cases, corresponding to the visual semantic features of EEG and channel-wise channel intensity features. Our purpose is to verify the effectiveness of our closed-loop iterative framework without caring about the performance of the above feature extractor itself. The encoder encodes the EEG generated by brain activity into semantic representations or energy features.
>
> Therefore, we can determine whether the current visual stimulation is a better stimulation by calculating the similarity score between the EEG corresponding features generated by the visual stimulation and the target features. From the control results of different channels in Figure 4, it can be seen that the EEG generated by the visual stimulation we designed is very close to the target EEG, reflecting the similarity of the brain activities of the two. Our framework provides guidance for online closed-loop neural control experiments on real subjects.

---

> > ### Author Response · Authors · 2024-11-21
> > **Rebuttal by authors**
> >
> > **Q6: "If the answer to the previous question is via the EEG encoder, then how can the encoder capture your concerned neural activity? How does encoder perform? How will the selection of the encoder influence the result?"**
> >
> > A6: As mentioned in Q1, a good encoder can obtain more effective representation of EEG activity, but our framework does not care about the type of EEG encoder. The contribution of this paper is to propose a new framework that introduces natural priors, which makes it possible to design optimal stimuli to regulate brain activity. Therefore, effective features will make the performance of our framework grow faster. Future research can focus on how to select effective embedding encoders to represent specific types of neural activities and further promote the application of this framework.
> >
> > **Q7: "What is the reward in the MDP?"**
> >
> > A7: For this question, please refer to Q3 in Global Response.
> >
> > **Q8: "For Figure 3.B, why do you choose subject 8 for demonstration? It seems the confidence interval is large. I wonder whether the similarity increase can pass the significance test."**
> >
> > A8: Due to limited space in the main body, we present the relevant results only for Subject 8. We report the complete performance of all subjects in the updated Appendix. Please review our updated pdf for more details.
> >
> > **Q9: "How to interpret the spectrograms in Figure 4.C? I can't see the difference or some trends from the figure."**
> >
> > A9: As you said, the changes in the spectrogram in Figure 4.C of our manuscript are not intuitive. We consider that this is mainly due to the performance of the encoding model and the fact that our framework relies heavily on the image feature similarity of CLIP. Although it is difficult to see the difference in the frequency-time diagram of the synthetic EEG data, the trend of the stimulus image approaching the target image can still be clearly perceived through the changes in the stimulus image at the bottom of Figure 4.C.
> >
> > **Q10: "How is Figure 4.D obtained? Why does the "random" also look so good?"**
> >
> > A10: The random EEG and target in Figure 4 of our manuscript also overlap a lot. This is because we use the encoding model instead of the real data of the brain, so it is limited by the prediction performance of the model itself. However, improving the encoding model is not the purpose of this article. In order to prove the validity of our framework, we added verification on the validity of the encoding model in the Appendix.
> >
> > In addition, Figure 4 of the work of Gifford et al. [1] shows that using the encoding model built end-to-end with CNN, the prediction accuracy of the zero-shot decreases rapidly from 0.4s onwards, which may be induced by visual stimulation. The real brain response has been basically completed, so the prediction results after nearly 100 data points are basically unreliable. Future work can continue to improve the performance of the encoding model on zero-shot EEG prediction, which will significantly improve the performance of our framework.
> >
> > References
> >
> > [1] Gifford A T, Dwivedi K, Roig G, et al. A large and rich EEG dataset for modeling human visual object recognition[J]. NeuroImage, 2022, 264: 119754.

---

> > > ### Comment · Reviewer_4DNY · 2024-11-22
> > > **Thanks for the revision**
> > >
> > > Thanks for the authors' effort in answering my questions and revising the paper. I have read the revised paper thoroughly and it's much better. However, I cannot find the original manuscript so I'm not completely sure how much the paper differs from the previous version unless the authors highlight the change, and it looks like a brand new paper to me. Therefore, I suggest that the paper may need going through another round of review to get a fairer judgement and will keep my decision.

---

> ### Author Response · Authors · 2024-11-23
> **Response for revision**
>
> Thank you for taking the time to carefully review our revised manuscript and for providing such valuable feedback. Your suggestions have significantly improved the quality of our work.
>
> We understand your concern about the substantial changes between the original and revised manuscripts. We have provided a detailed comparison of the two versions in the **Q4 section** of our **Global Response**. We encourage you to review this comparison to gain a better understanding of the specific modifications we have made.
>
> We hope these revisions have successfully addressed at least most of the concerns raised in your previous review. Thank you again for your time and consideration.

---

> ### Author Response · Authors · 2024-12-03
> **Response to Reviewer 4DNY**
>
> Dear reviewer,
>
> We are grateful for the time you've taken to carefully read our revision and provide valuable feedback. In order to provide greater clarity regarding the changes we made, we have updated the list of modifications in the **Global Paper Revision**. We hope this will facilitate a more efficient comparison between the two versions.
>
> The ICLR committee has kindly granted us three weeks to revise our manuscript, a valuable opportunity to further improve its quality. We are also pleased to note that both reviewers EgtU and FvdH have acknowledged the significant improvements made in the revised version. We respectfully request that you reevaluate our manuscript in light of these revisions.
>
> We are more than willing to address any remaining concerns or questions you may have within the allotted timeframe.
>
> Thank you for your time again.

---

### Official Review · Reviewer_EgtU · 2024-11-02

**Soundness:** 3
**Presentation:** 2
**Contribution:** 3
**Rating:** 6
**Confidence:** 4

**Summary:**

The authors proposed a closed-loop stimulation framework for EEG-based visual encoding, aiming to generate visual stimuli to elicit specific neural activities through controllable image generation strategy. In this framework, the authors control the stimulus image generation by approximating the brain activity evoked by the visual stimulation towards the desired neural response that corresponds to the candidate images rated by human users iteratively. Controlling visual stimuli in visual encoding studies is very important. Meanwhile, the stimulus images in most prior studies are relatively arbitrary as there is no standard criteria. The proposed framework provides a possible solution to this problem.

**Strengths:**

The proposed closed-loop framework for synthetic visual stimuli generation in novel in several ways, in terms of the retrieval strategy for identifying candidate images, the feature selection approach, and the method to addressing the problem of unknown target query image. The framework and related methodologies are well designed and presented in general.

**Weaknesses:**

The weaknesses of the manuscript lie in the lack of details and validations. For example, the details of encoding model are not sufficient. The authors described the architecture of the encoding model, however, the details for training such an encoding model is missing. he authors should provide details about training procedure, including data sources. Was the encoding model trained using data from multiple subjects or was it subject-specific? What is the method to validate the encoding model? More importantly, the encoding model was not adequately validated (at least I didn’t see any results related to the encoding model) given its critical role in the framework. In addition, what are the criteria and how to validate that the synthetic images are the “optimal” subset that can evoke specific neural activities? Similar issues exist in other modules, e.g., feature selection and interactive search. The authors are encouraged to validate each module separately rather than integratively.

**Questions:**

1. The limitations of past studies on closed-loop neural encoding/decoding were not adequately justified, weakening the contribution of the study.
2. The subtitles are not well match the items in the framework, making the manuscript is not easy to follow.
3. The encoding model has not been adequately validated. This module is very critical for the proposed framework. In addition, the implement details of the encoding model are not clear, e.g., was the model trained using individual data or data from multiple subject? How many training samples are used to train the encoding model? How to validate the model?
4. Is the EEG encoder which has been aligned with CLIP image features a good choice? This alignment may introduce bias in feature representation of the target and generated EEG signals. Why not a naive EEG encoder?
5. All the figures and tables are not referenced in the main text, making it quite difficult to read the figures. For example, what is encoded by the dot size in Figure 3c? What is the image with red boundary in Fig. 3d step 10?
4. Are there any failure cases? What I can imagine includes: 1) the random samples in the first round roulette wheel fail to cover the target; 2) The generated images at a certain iteration fail to cover the target. The authors are encouraged to discuss this issue.
6. “Since different stimulus images in our framework can produce the same or similar EEG features”—this could attribute to the existence of Metamers. However, other factors can not be overlooked: 1) the limitation of EEG (low spatial resolution) in quantifying brain activity. It might be possible that different stimulus image evoke similar EEG responses due to the limitations of EEG. 2) The limitation of the model for EEG feature prediction (the encoding model 3.1). The authors are encouraged to make justifications more carefully.

Other issues:
“quotation marks” are not in right format
The font size of some text in the figures are too small to read.
Typos: in Figure 4 captions: (F) is for O2 channel?

---

> ### Author Response · Authors · 2024-11-21
> **Rebuttal by authors**
>
> Thank you for your thorough analysis and constructive feedback on our paper. Here are our point-by-point responses to your questions:
>
> **Q1: "....What are the criteria and how to validate that the synthetic images are the “optimal” subset that can evoke specific neural activities? Similar issues exist in other modules, e.g., feature selection and interactive search. "**
>
> A1: We have explained it in detail in In Section 3.1 Closed-loop Framework. We determine whether the current image is close to the optimal stimulus by calculating the similarity score between the EEG features generated by the current stimulus and the target stimulus image. In addition, in order to avoid misunderstanding, we modified the section on "Feature Selection" in our original manuscript. Seciton 3.1 is now the design of the entire framework, including the definition of the EEG features approximated in this paper. Among them, "Interactive Search" is a case of EEG semantic representation in our closed-loop framework, which is more suitable for solving with an iterative algorithm based on retrieval rather than generation. We have carefully verified the effectiveness of each module. Please refer to the updated PDF for more details.
>
> **Q2: "...The limitations of past studies on closed-loop neural encoding/decoding were not adequately justified, weakening the contribution of the study."**
>
> A2: Thank you for your insight thoughts on this research. Most of the past research has stayed at the level of encoding and decoding methods of different modal neural data or behavioral data, but there are few studies that combine the characteristics of the two technologies to explore potential applications. Researches like Tolias and DiCarlo et al. on mice and monkeys confirmed that specific images can stimulate the activity of target neurons [1][2][3]. Luo et al.'s research on visual cortex selectivity revealed the potential of combined encoding/decoding research [4][5]. Based on previous research, this paper provides a closed-loop stimulus generation framework that introduces natural priors to further explore the potential of optimally designed stimuli in regulating brain activity. In Appendix A.1 and A.2 of the experiment, we give the results of the verification of the effectiveness of the encoding model and decoder to support the methodology we proposed. We strongly agree with your suggestions, and our future work will focus on exploring the limitations of closed-loop neural encoding/decoding to provide more support for this methodology.
>
> **Q3: "...The subtitles are not well match the items in the framework, making the manuscript is not easy to follow."**
>
> A3: According to your suggestion, we have updated the manuscript to better align the subtitles with the items in the framework, making it easier to follow. Additionally, the general writing improvements are addressed in the Global Response. See the updated pdf for more revised results.
>
> **Q4: "...This module is very critical for the proposed framework. In addition, the implement details of the encoding model are not clear, e.g., was the model trained using individual data or data from multiple subject? How many training samples are used to train the encoding model? How to validate the model ?"**
>
> A4: To address your concerns, we have added verification of the effectiveness of the encoding model in Appendix A.1 and A.2. We have supplemented the detailed experimental configuration of the encoding model in Section 4.1. Due to the non-stationarity of EEG and differences between subjects, all our models are in-subject. The THINGS-EEG2 dataset is used to train, test, and validate our encoding model. The dataset includes paired visual and evoked EEG data from 10 different subjects. For each subject, the training set contains 1,654 different visual stimulus objects (10 images per object) and the corresponding EEG responses. The test set contains 200x12 image samples and 200x1 EEG data. Detailed information on the specific definition of the black box encoding model is provided in Section 3.1, please refer to the updated pdf for more information.
>
> **Q5: "Is the EEG encoder which has been aligned with CLIP image features a good choice? This alignment may introduce bias in feature representation of the target and generated EEG signals. Why not a naive EEG encoder ?"**
>
> A5: The EEG Encoder in our Figure 1 is a general representation, and its specific structure depends on the type of EEG feature case to be regulated. For the case of EEG semantic features in Section 3.3, the EEG feature extractor uses the EEG Encoder that has been aligned with the CLIP image features. However, for the case of channel-wise channel intensity features in Section 3.4, the EEG feature extractor calculates the PSD features of different channels. Therefore, this EEG Encoder can be a feature predictor that aligns arbitrary features, not limited to alignment with images or channel energy.

---

> ### Author Response · Authors · 2024-11-21
> **Rebuttal by authors**
>
> **Q6: "All the figures and tables are not referenced in the main text, making it quite difficult to read the figures...."**
>
> A6: We have updated the figure captions and text to clarify these points, ensuring that the figures are fully understood in the context of the manuscript. Together, Figures 3c and 3d demonstrate the effectiveness of our framework in converging on images that match the target's neural representation through iterative refinement.
>
> **Q7: "Are there any failure cases? What I can imagine includes: 1) the random samples in the first round roulette wheel fail to cover the target; 2) The generated images at a certain iteration fail to cover the target. The authors are encouraged to discuss this issue."**
>
> A7: This is an interesting question. We considered this problem when designing the algorithm. Failure is certain to occur. Therefore, we also used a variety of means in the specific implementation process to ensure the diversity of the candidate sample set and ensure that the optimization will not continue to fall into Goal Drift.
>
> First, assuming the case of regulating the semantic representation of EEG under the retrieval task, if a round of random samples fails to cover the target, a new round of samples will be obtained through cumulative probability sampling. Since the first round of random sampling is uniformly distributed in space, no matter where the target is in the feature space, our algorithm always guides the samples taken out in the next iteration to move closer to the target direction. Even if the stimulation sample moves in the wrong direction in a certain iteration, the sampling probability of the wrong direction sample will be reduced, and the sampling direction will be gradually readjusted.
>
> On the other hand, if the case is to regulate the strength of brain channels under the generation task, we adopt crossover and mutation operations at the image feature level to increase the diversity of the iterative population. At the same time, each round of iteration selects individuals with high fitness and retains them as candidate stimuli for the next round. After adding the genetic algorithm, our framework can help the generation model understand which stimulus image features are the preferences of the target EEG features, and guide the generation model to continuously update the image details in this direction.
>
>
>
> **Q8: "...other factors can not be overlooked: 1) the limitation of EEG (low spatial resolution) in quantifying brain activity. It might be possible that different stimulus image evoke similar EEG responses due to the limitations of EEG. 2) The limitation of the model for EEG feature prediction (the encoding model 3.1)..."**
>
> A8: Thank you for your suggestion, we agree with your point of view. We chose EEG because of its low online acquisition cost and timely feedback from subjects. However, EEG signals are non-stationary and are greatly affected by factors such as equipment, environment, and psychological state of subjects. Therefore, there may be problems with inaccurate control of specific channels by stimulus images, so the real-time performance of the system is particularly important. Our encoder determines the feature type of EEG response, so even if metamers exist, the goal of approaching the target EEG response feature can still be achieved. In addition, since the performance of the EEG encoding model has a great impact on our framework, we supplemented the experiments in the Appendix to demonstrate the effectiveness of the encoding model used in the experiment.
>
> **Q9: "...'quotation marks' are not in right format The font size of some text in the figures are too small to read. "**
>
> A9: Thank you for your suggestion, we have updated the manuscript to ensure that quotes are formatted correctly and to improve font size for better readability.
>
> **Q10: "Typos: in Figure 4 captions: (F) is for O2 channel ?"**
>
> A10: Yes, Figure 4F represents the O2 channel. We have fixed this issue in the manuscript.
>
>
> References
>
> [1] Walker E Y, Sinz F H, Cobos E, et al. Inception loops discover what excites neurons most using deep predictive models[J]. Nature neuroscience, 2019, 22(12): 2060-2065.
>
> [2] Bashivan P, Kar K, DiCarlo J J. Neural population control via deep image synthesis[J]. Science, 2019, 364(6439): eaav9436.
>
> [3] Pierzchlewicz P, Willeke K, Nix A, et al. Energy guided diffusion for generating neurally exciting images[J]. Advances in Neural Information Processing Systems, 2024, 36.
>
> [4] Luo A, Henderson M M, Tarr M J, et al. BrainSCUBA: Fine-Grained Natural Language Captions of Visual Cortex Selectivity[C]//The Twelfth International Conference on Learning Representations.
>
> [5] Luo A, Henderson M, Wehbe L, et al. Brain diffusion for visual exploration: Cortical discovery using large scale generative models[J]. Advances in Neural Information Processing Systems, 2024, 36.

---

> > ### Comment · Reviewer_EgtU · 2024-11-25
> >
> > I would like to thank the authors for addressing my concerns. They have made substantial revisions such that it looks like a new submission to me. Most of my concerns have been addressed in the revision, however, I am not sure if this is the true spirit of rebuttal.

---

> > > ### Author Response · Authors · 2024-11-28
> > > **Response for revision**
> > >
> > > We appreciate your consideration. ICLR has allocated a three-week period for rebuttal and discussion, allowing ample time for the reviewers' feedback to be thoroughly addressed and for the manuscript to be improved accordingly. Therefore, our revisions are aligned with the spirit and guidelines of ICLR.

---

### Official Review · Reviewer_CRyH · 2024-11-03

**Soundness:** 4
**Presentation:** 3
**Contribution:** 4
**Rating:** 8
**Confidence:** 3

**Summary:**

This is a highly innovative study demonstrating the capability to identify visual stimuli that closely match the original stimuli eliciting specific EEG activity patterns. The algorithm is well-explained and, to my knowledge, represents one of the first successful applications of this approach with EEG data.

**Strengths:**

Very interesting study, timely, solves an important question, is generalizable.

**Weaknesses:**

I can hardly identify any significant limitations in the current study. However, I have two questions:

The authors state that the identified stimulus is "optimal." Based on the MDP formulation of the algorithm, I understand that it finds a local minimum. Could you clarify how this approach ensures finding a global optimum, rather than a local one?

Why did you limit the comparison to the first 250 ms (Figure 4D)? While the initial 250 ms may indeed capture critical visual information, it is common in EEG analysis to display the full 1000 ms post-stimulus data. Could you elaborate on this choice?

**Questions:**

The authors state that the identified stimulus is "optimal." Based on the MDP formulation of the algorithm, I understand that it finds a local minimum. Could you clarify how this approach ensures finding a global optimum, rather than a local one?

Why did you limit the comparison to the first 250 ms (Figure 4D)? While the initial 250 ms may indeed capture critical visual information, it is common in EEG analysis to display the full 1000 ms post-stimulus data. Could you elaborate on this choice?

---

> ### Author Response · Authors · 2024-11-21
> **Rebuttal by authors**
>
> Thank you for your recognition of our work. Here are our responses to your questions:
>
> **Q1: The authors state that the identified stimulus is "optimal." Based on the MDP formulation of the algorithm, I understand that it finds a local minimum. Could you clarify how this approach ensures finding a global optimum, rather than a local one?**
>
> A1: As you correctly point out, using the Markov Decision Process (MDP) formulation, it is possible for the algorithm to converge to a local optimum in the retrieval case, especially in non-convex search spaces. However, in our generative case, there is no real global optimum. Human visual perception is a complex system influenced by many factors, introducing significant randomness, which makes it challenging to define convex optimization problems with precise mathematical formulations.
>
> Therefore, we adopt heuristic methods (such as genetic algorithms) to determine the optimal results. For more detailed details on the methods in the manuscript please see **Section 3** in the updated pdf. For the generation task, we introduce random sampling in each iteration. Regarding the termination conditions, (1) if the increment of similarity between features is less than 10e-4, we consider the process to have converged, and (2) if the number of iterations reaches 90, the process stops. As shown in **Figure 4A**, the similarity between brain activity features increases with each iteration and tends to be stable. In addition, the example in **Figure 4C** shows that the optimal visual stimulation is consistent with human prior knowledge. In addition, if we relax the iteration limit and allow more iterations, the generation model may continue to optimize until it reaches the upper limit of "optimality", because additional iterations provide the algorithm with more opportunities to explore and improve the solution.
>
> **Q2: Why did you limit the comparison to the first 250 ms (Figure 4D)? While the initial 250 ms may indeed capture critical visual information, it is common in EEG analysis to display the full 1000 ms post-stimulus data. Could you elaborate on this choice?**
>
> A2: Thank you for pointing this out. As shown in **Figures 4D, 4E, and 4F**, the value of 250 refers to the number of data points, not milliseconds, which means that we are showing data within 1000ms at a sampling rate of 250Hz. We realize that this may be confusing. To improve clarity, we have updated the unit to milliseconds (ms). The dataset we used, THINGS-EEG2, originally spanned 1000 ms and had a sampling rate of 1000 Hz. During preprocessing, the following steps were applied using Matlab (R2020b) and the EEGlab (v14.0.0b) toolbox as described in the dataset publication: (1) the data were filtered using a Hamming window FIR filter with a 0.1 Hz high-pass and 100 Hz low-pass filter, (2) the data were re-referenced to the mean reference and downsampled to 250 Hz [1].
>
>
> References
>
> [1] Gifford A T, Dwivedi K, Roig G, et al. A large and rich EEG dataset for modeling human visual object recognition[J]. NeuroImage, 2022, 264: 119754.

---

> > ### Comment · Reviewer_CRyH · 2024-11-25
> >
> > thanks for the clarification on Q2 and the discussion on Q1. I am not sure if I agree with our discussion of Q1. Here I meant an algorithmic/computational proof of optimality, but I also understand that this might be beyond the focus of the current paper. My suggestion is to make this more clear in your paper to avoid confusion. I keep my score as it is, this is a very nice paper and I will be happy to see it in ICLR.

---

### Official Review · Reviewer_FvdH · 2024-11-04

**Soundness:** 2
**Presentation:** 2
**Contribution:** 3
**Rating:** 6
**Confidence:** 2

**Summary:**

This paper develops a method for choosing the optimal image stimulus to present to a human subject to elicit a specific desired pattern of neural activity (as measured using EEG).

**Strengths:**

The problem that this paper takes on is very interesting. I am aware of previous research that has attempted to find preferred visual stimuli for single neurons, so as to figure out what that neuron "prefers", but this paper seems to be taking on a related but quite different issue, which is: given a whole pattern of population activity, what stimulus would elicit that overall pattern? This seems like a project that may have useful clinical applications in the future, as well as being scientifically interesting in its own right.

**Weaknesses:**

I found the paper hard to follow. I admit that a contributing factor here may be my own lack of experience with respect to some of the techniques the paper uses, such as EEG data, diffusion models, and genetic algorithms. However, I do think that the presentation of the paper could be much clearer, and I will list some examples below of specific issues that came up with respect to clarity.

- Most of the figures I did not understand, and as far as I could tell, the figures aren't referred to in the main text, so it was difficult to situate what the purpose of each figure was in the overall narrative of the paper.
- It is unclear what the purpose of the MDP is in Section 3.2 (see Questions below).

It would probably have been useful to include a Supplemental section to explain some of the methods in more detail.

**Questions:**

In Sec 3.2, what do the actions and states of the MDP refer to in this context? Are the actions features, because the algorithm is selecting features of the neural activity to represent? Or are the actions the selected images to be used as visual stimuli?

What is the motivation for not updating the gradients in the model? The abstract says this allows "us to directly analyze the relationship between the administered visual stimuli and the targeted brain activity", but I wasn't sure why this is the case or where in the paper this motivation is fully explained or justified.

In Figure 1, what is the difference between "selection" and "action"?
In Fig 2, the distance metric seems to be applied to images, but I thought the point was to compare induced and target neural activities.

---

> ### Author Response · Authors · 2024-11-21
> **Rebuttal by authors**
>
> Thank you for your thoughtful review and for recognizing the importance of our work. The following is a point-by-point  response to your review:
>
> **Q1: ".. the figures aren't referred to in the main text, so it was difficult to situate what the purpose of each figure was in the overall narrative of the paper."**
>
> A: We have revised the caption of all figures you mentioned to clarify their purpose and improve their coherence with the overall structure of the paper. In addition, we have made corresponding changes to the paper to ensure that each figure is appropriately cited and more effectively supports the arguments of the paper. Please refer to the Global Response for the manuscript and see our updated PDF file.
>
>
> **Q2: "It is unclear what the purpose of the MDP is in Section 3.2 (see Questions below)."**
>
> A: Our closed-loop iterative algorithm based on similarity score feedback is similar to MDP. For more details, please refer to Section 3 of our uploaded latest pdf file.
>
> **Q3: "In Sec 3.2, what do the actions and states of the MDP refer to in this context? Are the actions features, because the algorithm is selecting features of the neural activity to represent? Or are the actions the selected images to be used as visual stimuli?"**
>
> A: Please see Section 3 of the latest PDF. The state s can be represented as the probability distribution of each image P(u) in the image database belonging to the target category. The state updated after each iteration can be regarded as a state transition in MDP. In each iteration, our framework needs to decide which image to be selected. This can be viewed as an action in the MDP.
>
>
> **Q4: "What is the motivation for not updating the gradients in the model? ...... I wasn't sure why this is the case or where in the paper this motivation is fully explained or justified."**
>
> A: Please refer to the Q1 of Global Response for the content about the motivation of this work. Our framework is designed with three core pre-trained components: the black-box model, feature extractor, and the downstream tasks, which include an image retrieval task for regulating semantic representation and an image generation task for regulating channel-wise spectrum feature. These components are pre-trained specifically for different subjects and, therefore, do not require training during our closed-loop iterations. This iterative approach focuses solely on refining the stimulus without modifying model parameters, allowing us to assess the direct impact of visual stimuli on brain activity without introducing variability from gradient updates.
>
>
> **Q5: "In Figure 1, what is the difference between "selection" and "action"?"**
>
> A: Thank you for pointing out this point we did not clarify. To avoid misunderstanding, we have changed the "Selection" in the Figure 1 to "Preference". Our framework is expected to be able to identify which image features (color, texture, background, etc.) are preferred according to the feedback of target EEG features, while the "action" involves deciding the next round of images according to the similarity score based on these similarity evaluations.
>
> In the retrieval task, given that the exact query image is unknown, we initialize with a random set of images and iteratively narrow down to those with higher semantic similarity to the target. Therefore, the "action" refers to choosing which images in each iteration based on their similarity to the target brain features to construct the next stimulus set. The system progressively learns which features are most relevant to the target class by tracking similarities across iterations, assigning higher weights to these features in future steps.
>
> In the generation task, "action" involves identifying images with high similarity to the target, where images with high similarity scores are kept and used as a basis for generating new samples. Then applying a genetic algorithm to cross and mutate these images, while ensuring they retain coherent, human-recognizable semantic content.
>
> **Q6: "In Fig 2, the distance metric seems to be applied to images, but I thought the point was to compare induced and target neural activities."**
>
> Thank you for pointing out this key information. We have improved Figure 2 to make the details of our framework more clear. The distance metric in Figure 2 is indeed derived from the distance between images. Our hypothesis is that this similarity between images maps the similarity between the induced and target neural activities (although the existence of Metamers is not excluded), allowing us to use image features as a proxy for brain activity. Even if this mapping relationship is intractable, we can still perform a heuristic solution in a gradient-free way. Therefore, we use the visual features of the image embedded in the feature space to approximate the desired target image, thereby continuously bridging the gap between the current stimulus image and the target brain activity.

---

> ### Comment · Reviewer_FvdH · 2024-11-26
>
> I thank the authors for their responses to my questions. I've read through the revised PDF. I think this newer version of the paper is clearer than the previous version, but still feel there are several points where things are not very clear. Most figures are referred to in the main text, but not Figure 2, which is still confusing to look at (at least for me), and also not referred to directly by the main text.  The main text of Section 3 I also find very hard to follow - even though it seems to provide more details, it is hard to parse. For example, I wasn't sure where the target category comes from - is this a category of the image, like dog vs cat? I thought the target was a specific neural activity pattern.
>
> I appreciate the details of the genetic algorithm, which I don't think were given in the previous iteration of the paper, but also wish more motivation and intuition were given for why this genetic algorithm was chosen to generate new images as opposed to some other algorithm.
>
> Overall, I will keep my rating as is because I think the paper still needs to be substantially clearer (although to be clear, I do think this is better than the previous version, and my only reason for keeping the rating the same is that my decision as to whether it is ready to be published in the current state is unchanged). I would like to see this paper published eventually, but just think that it needs further revision into a version that is much clearer and easier to follow.

---

> ### Author Response · Authors · 2024-11-27
> **Rebuttal by authors**
>
> We appreciate your thorough review and valuable suggestions on our revised manuscript. Below are some of the changes we made and our point-by-point responses to your questions:
>
> **Q1: “...Most figures are referred to in the main text, but not Figure 2, which is still confusing to look at (at least for me), and also not referred to directly by the main text. ”**
>
> A: To help readers with limited background knowledge in this field to more easily understand our pipeline, we have revised the caption of Figure 2. And we have cited the **Figure 2** in **Section 3**, line 142 of the main text.
>
> **Q2: “The main text of Section 3 I also find very hard to follow - even though it seems to provide more details, it is hard to parse. For example, I wasn't sure where the target category comes from - is this a category of the image, like dog vs cat? I thought the target was a specific neural activity pattern.”**
>
> A: Our target refers to a specific neural activity pattern that we want to evoke in the brain. This neural pattern is from evoked or predicted EEG signals while presenting certain images, which we determined in advance as representative of the target brain activity. Regarding the “target category”, our previous description may have caused some confusion, so we revised the explanation of the way to obtain the target EEG features in **Section 4.1**.
>
> In other words, the target is an EEG feature that corresponds to the brain's response when viewing a particular image. We use this target EEG feature to search for other images that can generate brain activity patterns most similar to those evoked by the target image. These images, which are chosen based on their ability to induce similar neural activity patterns, form the core of our study.
>
> **Q3: “I appreciate the details of the genetic algorithm, which I don't think were given in the previous iteration of the paper, but also wish more motivation and intuition were given for why this genetic algorithm was chosen to generate new images as opposed to some other algorithm.”**
>
> A: The relationship between image features and brain activity is inherently nonlinear, presenting significant challenges for direct modeling. As for using the Genetic Algorithm (GA) during the generation phase, we have two main considerations.
>
> - First, GA can effectively explore the expansive and complex image search space through population diversity, which mitigates the risk of converging to local optima. This ensures the identification of a solution that closely approximates the target EEG pattern.
>
> - Second, the GA is particularly well-suited for high-dimensional, complex optimization problems, such as ours. Unlike gradient-based methods, GA does not rely on gradient information, making it more appropriate for our black-box model-driven framework.
>
> Specifically, GA can iteratively refine the image currently presented to the system via a process inspired by fitness—defined as the similarity between the target EEG and the brain activity elicited by the image. This iterative optimization enables the identification of images that are more likely to provoke the desired neural response in a computationally efficient manner. These images can then be fed into the generation model for further editing, yielding higher-fitness samples that better align with the target neural patterns.
>
> However, there are challenges associated with the use of GA, including excessive iterations, slow convergence, and relatively low stability. Given the maturity of the field of evolutionary computing, future research could focus on leveraging more efficient heuristic algorithms to address these limitations and enhance the overall optimization process.
>
> **Q4: “ ...my only reason for keeping the rating the same is that my decision as to whether it is ready to be published in the current state is unchanged). I would like to see this paper published eventually, but just think that it needs further revision into a version that is much clearer and easier to follow.”**
>
> A: Thank you for your suggestions. We have added new annotations to **Figure 1**, hoping they will help clarify our whole framework. Additionally, we have polished **Section 3 Method** in the manuscript accordingly to make it more understandable and logical.

---

> > ### Author Response · Authors · 2024-12-03
> > **Response to Reviewer FvdH**
> >
> > Sorry to bother you. We are about to run out of time to respond.
> >
> > We have made complements and explanations for this work with the help of all the reviews. We would be grateful if you could confirm whether the rebuttal meets your expectations and if there is any other suggestion.
> >
> > Thank you once again for your time and insightful comments!

---

> > > ### Comment · Reviewer_FvdH · 2024-12-03
> > >
> > > I apologize for my delayed response to the authors. I appreciate that they have responded to each of my comments and made efforts to make the paper clearer in all the ways suggested, and will increase my score accordingly.

---

> ### Author Response · Authors · 2024-11-28
> **Response to revision**
>
> Dear Reviewer,
>
> We kindly request your feedback on whether our response, along with the revised manuscript, sufficiently addresses your concerns. Thank you again for your time and consideration.

---

### Author Response · Authors · 2024-11-21
**Global Response**

We are sincerely grateful to the reviewers for dedicating their time and effort to review our work. We are excited about the consensus among the reviewers regarding the novelty (FvdH, CRyH, EgtU, 4DNY) and significance (FvdH, 4DNY) of our work. Unfortunately, our past submission has been plagued by clerical errors and lack of clarity. So based on suggestions from reviewers, we have conducted a comprehensive revision of our manuscript to clarify the methodology, add missing technical details and polish our paper. We hope to do our best to clarify what we did not explain clearly in the manuscript to allay the concerns of the reviewers. In the following, we provide a Global Response to the reviewers' questions and concerns.

**Q1: What is the motivation of this work, particularly with regard to the establishment of the closed-loop system and black-box modeling?**

A1: In a closed-loop feedback system, brain activity is influenced by numerous factors, making it highly susceptible to interference. As a result, EEG signals recorded from a single stimulus may not fully capture the desired brain response. This necessitates multiple rounds of stimulation to reinforce and stabilize the response. As mentioned in the Introduction, it has been seen as an significant technique to focus on closed-loop regulation, both at the level of individual neurons and in larger-scale EEG systems. However, these studies have some limitations, such as lack of natural priors and generalization which human beings can understand. This underscores the necessity of the framework we propose.

The black-box modeling we show in **Figure 1** is to verify the effectiveness of our training-free closed-loop framework. We assume that our framework still work even the specific structure of the encoding model is unknown. This can help us ignore the gains brought by the encoding model and focus on the advancement of the framework itself. So we use a simple encoding model with frozen weights, without the calculation of gradients. Given that the human brain does not provide direct access to neuronal activation patterns or the sequence of bioelectrical responses, these processes are not easily derivable. Therefore, in order to more generally simulate the specific process of our method in regulating human brain responses, it becomes necessary and meaningful to introduce a black-box model.

**Q2: How is the EEG encoding model trained to ensure the reliable synthesis of EEG?**

A2: In our framework, the validity and diversity of EEG signals are prioritized over strict reliability. The framework aims to regulate target-specific neural activity of individuals. The primary requirement for our pretrained encoding model is to encode natural image to EEG, to approximate the brain activity in response to visual perception. Previous studies have shown that EEG signals evoked by visual stimuli can be aligned with stimuli, which can be effectively modeled for downstream tasks such as decoding and reconstruction.

**Q3: How is the MDP algorithm implemented in this work, including a detailed explanation of the actions, rewards, and other relevant components?**

A3: Our closed-loop stimulus generation framework only depends on the current state and the current action (the current image search space), which has uncertainty and randomness in the search space. At the same time, the feedback of the action (selecting the stimulus image) and the reward (similarity score) is clear, so we model the entire iterative process through MDP. When performing retrieval as a setting for finding visual stimuli, the space of states (current stimulus sequence) and actions (selecting stimulus) is explicitly limited, that is, the complexity of the problem is controllable. When using a generative model and combining iterative generation of visual stimuli with evolutionary computation, we set the population fitness (similarity score threshold) as the condition for terminating evolution, so that the space of actions is also limited in this framework.

**Q4: Figures and tables are not referenced, making it quite difficult to read the figures. The presentation lacks clarity in certain areas, making the manuscript is not easy to follow.**

A4: Below, we outline the key changes made in response to your comments. A modified version of the manuscript for further clarification has been uploaded in the **PDF**.

(1) We have revised the full manuscript, unified the mathematical notation, and refered each figure and table in a logical manner. Moreover, we have supplemented with more implementation details (**Section 4**).

(2) We have updated the **Figure 1**, **Figure 2** and their captions to make our entire framework more clear.

(3) In **Section 3.3** and **Section 3.4**, we have updated the description of the algorithm (or pseudocode) in two settings (retrieval and generation) to for better understanding to our implementation details and framework.

---

### Author Response · Authors · 2024-11-26
**Paper revision**

We would like to express our sincere gratitude to the reviewers for their insightful and valuable comments on our manuscript. Based on the feedback, we have made several updates to improve the clarity and quality of the writing, as well as to better support the experimental results.

To facilitate the comparison between the two versions, we have listed the changes below:

**1. General Revisions**: In response to the written errors and unclear expressions raised, we have revised the manuscript to improve its clarity and readability. Specifically, we have corrected the mathematical notations, added the correct references for each figure and table, and provided additional detailed implementation information in **Section 4**.

**2. Figures and Descriptions**:

- **Figure 1**: For the extracted features section in our framework, we have updated the caption to provide a more comprehensive illustration. This revision offers a clearer understanding of our motivation.

- **Figure 2**: We have updated **Figure 2**, which offers a clearer and more concise presentation. On the left, the figure provides a step-by-step breakdown of the process, with more explicit descriptions of each step in the framework. On the right, it includes a detailed illustration of the pipeline, followed by descriptions of the two case studies.

**3. Section 3 Reorganization**:

We have reorganized the **Section 3 "Method"** and created a new **subsection 3.1: "Closed-loop Framework"** to further clarify the mathematical formula, making the section more intuitive to readers. In **Sections 3.3 and 3.4**, we have updated the descriptions of the algorithms (or pseudocode) for both the retrieval and generation settings, which now include more detailed steps and improved clarity to better convey our implementation process.

**4. Appendix Updates**:

- **Appendix A.1**: More details has been added regarding the dataset, the interactive search process, and the heuristic generation algorithm, providing a complete insight into our methodology.  (**Page 13, FvdH**)

- **Appendix A.2**: We have included extensive quantitive results of all subjects in both cases. We have also provided performance comparisons of iterative improvement in different target images across all 10 subjects in EEG semantic representation and spectrum intensity (**Table A.1**). Additionally, we have reported the improvement through in-subject t-tests across all subjects for both cases and correlation analysis between EEG features and CLIP representations (**Figure A.3**). (**Page 15, 4DNY**)

- **Appendix A.3**: In this section, we elaborate on the validity verification of synthetic EEG signals. Specifically, we conducted MSE and Pearson's correlation coefficient on synthetic EEG signals from AlexNet and CORnet-S across all 10 subjects. We also present results from training with ATM-S on two DNN models with pre-trained end-to-end models and randomly initialized end-to-end models (**Table A.2, Figure A.4- Figure A.8**). All these assesments are able to comprehensively vertify the performance of EEG encoding model and support our experiment settings on pretrained end-to-end Alexnet. (**Page 17, 4DNY and EgtU**)

- **Appendix A.4**: We have added supplementary examples, including EEG semantic representation and intensity cases, where we show the iterative complete process of different subjects and targets in each case. Additionally, we have provided examples where regulation failure gradually leads to Goal Drift (**Figure A.4.2**). (**Page 23, EgtU**)

**Changs 1&2&3** have led the improvements of the manuscript without affecting the experimental results presented in this paper. **Changes 4** brings some validations.

---

### Meta-Review · Area_Chair_K625 · 2024-12-08

**Metareview:**

This work presents a closed loop methodology to identify optimal visual stimuli for maximally eliciting neural activation in the brain. The method relies on a combination of diffusion image generation and Markov-Decision-Process-like updates, iteratively generating new images and updating the latent representation of image-induced activity. The authors then test their data on pre-recorded EEG datasets, and use a neural network model trained to generate EEG data given a stimulus to test their framework. The reviewers were split, appreciating the general mathematical formalization of the method and goals of the paper. Several weaknesses were mentioned, with a large majority falling into the category of either clarity or insufficiency of the experimental validation. While the authors improved the clarity in new versions of the manuscript, there remains some question of how well this method will work in real applications, where a number of practical challenges of EEG stability/drift, changes in noise etc. can potentially reveal unforeseen instabilities in the method. Thus I believe a more thorough validation and, especially, either an experimentally validated test or proof that the simulated EEG suffices, would be key to a future submission.

**Additional Comments On Reviewer Discussion:**

There were a number of clarity issues that were discussed and seemingly clarified. One challenge that arose was that one reviewer was unclear on the exact changes in the updated manuscript. I suggest to the senior area chairs & above to consider "freezing" the original PDF and allowing the updated PDF to be uploaded, only replacing the original submission after the decisions have been made.

---

### Decision · Program_Chairs · 2025-01-22

Reject